# EDITMGT: UNLEASHING POTENTIALS OF MASKED GENERATIVE TRANSFORMERS IN IMAGE EDITING

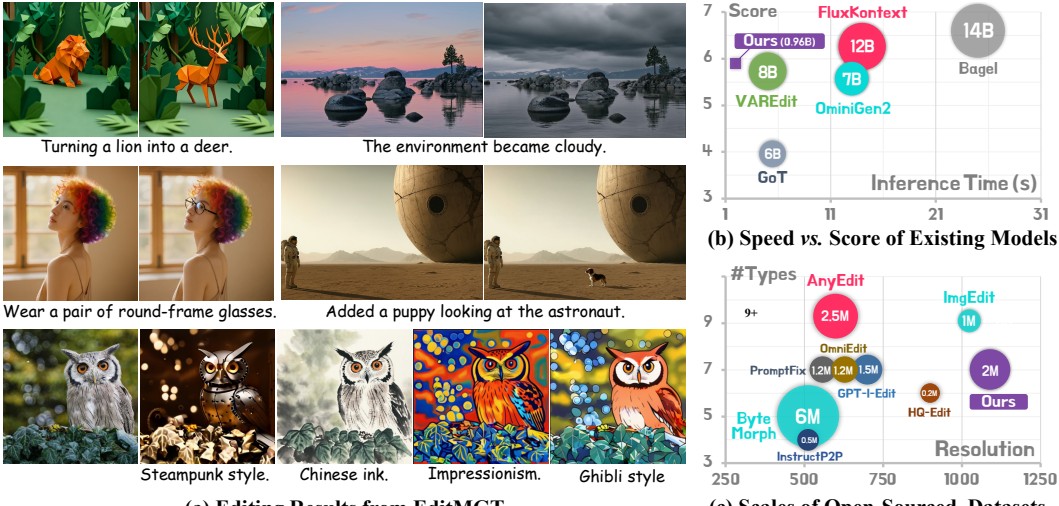

Figure 1: Overview of EditMGT and CrispEdit-2M. EditMGT, the first MGT-based model, performs editing in 2s with 960M parameters, $6\times$ faster than models of comparable performance; CrispEdit-2M provides 2M high-resolution ($\geq 1024$) editing samples spanning 7 distinct categories.

## ABSTRACT

Recent advances in diffusion models (DMs) have achieved exceptional visual quality in image editing tasks. However, the global denoising dynamics of DMs inherently conflate local editing targets with the full-image context, leading to unintended modifications in non-target regions. In this paper, we shift our attention beyond DMs and turn to Masked Generative Transformers (MGTs) as an alternative approach to tackle this challenge. By predicting multiple masked tokens rather than holistic refinement, MGTs exhibit a localized decoding paradigm that endows them with the inherent capacity to explicitly preserve non-relevant regions during the editing process. Building upon this insight, we introduce the first MGT-based image editing framework, termed EDITMGT. We first demonstrate that MGT's cross-attention maps provide informative localization signals for localizing edit-relevant regions and devise a *multi-layer attention consolidation* scheme that refines these maps to achieve fine-grained and precise localization. On top of these adaptive localization results, we introduce *region-hold sampling*, which restricts token flipping within low-attention areas to suppress spurious edits, thereby confining modifications to the intended target regions and preserving the integrity of surrounding non-target areas. To train EditMGT, we construct CrispEdit-2M, a high-resolution ($\geq 1024$) dataset spanning seven diverse editing categories. Without introducing additional parameters, we adapt a pre-trained text-to-image MGT into an image editing model through attention injection. Extensive experiments across four standard benchmarks demonstrate that, with fewer than 1B parameters, our model achieves state-of-the-art image similarity performance while enabling $6\times$ faster editing. Moreover, it delivers comparable or superior editing quality, with improvements of $3.6\%$ and $17.6\%$ on style change and style transfer tasks, respectively. ✪ More information can be found from the **Anonymous Page**: *https://anoy1314.github.io*.

# 1 INTRODUCTION

Image editing has witnessed remarkable progress in the era of generative artificial intelligence, shifting the paradigm from pure synthesis toward fine-grained interactive control (Banh & Strobel, 2023; Feuerriegel et al., 2024). The predominant paradigm in this domain is (DMs) (Croitoru et al., 2023; Hertz et al., 2022; Brooks et al., 2023), which achieve impressive visual fidelity through iterative denoising processes. However, this core mechanism introduces a critical limitation: the global nature of the denoising process frequently leads to unintended spurious edits, causing modifications to "leak" into regions that should remain unchanged (Hu et al., 2025; Mao et al., 2025).

Previous approaches have addressed this challenge through **three** primary paradigms: (1) leveraging large-scale, high-quality training data to enable models to implicitly learn such constraints (Yu et al., 2025); (2) employing manually predefined masks in conjunction with inpainting models (Zhang et al., 2024; Bai et al., 2024a); and (3) utilizing inversion techniques to establish mappings from non-edited regions to corresponding Gaussian noise subspaces (Mokady et al., 2023; Tang et al., 2024; Avrahami et al., 2022; Rout et al., 2024). The first approach cannot explicitly guarantee that irrelevant regions remain unmodified, while the second suffers from limited flexibility due to its dependence on pre-trained inpainting models. The third methodology exhibits slow inference speed and may still lead to unintended modifications (Mu et al., 2025; Hertz et al., 2022; Wu et al., 2025c).

To address these limitations, we turn our attention to an alternative paradigm—Masked Generative Transformers (MGTs). Unlike diffusion models that rely on iterative holistic refinement, MGTs synthesize images by predicting multiple masked tokens in parallel (Chang et al., 2022). This autoregressive formulation not only offers an efficient generation process, but also inherently supports zero-shot image inpainting with predefined masks (Patil et al., 2024), thereby fundamentally avoiding the entanglement issues of DMs and offering a natural mechanism to explicitly preserve non-target regions of the original image. Grounded in the intrinsic strengths of MGTs, we pinpoint two capabilities essential for effective image editing: ❶ *adaptive localization of edit-relevant regions* and ❷ *explicit preservation of non-relevant regions during inference*.

To this end, we propose **EDITMGT** in this paper, the first MGT-based image editing framework designed to fundamentally resolve the aforementioned editing leakage problem. Leveraging MGT's inherent local decoding property, our method can perform zero-shot model updates exclusively within specified editing regions (*e.g.*, user-provided masks) while ensuring complete preservation of edit-irrelevant areas by maintaining tokens in these regions entirely unmodified. Building upon this foundation, we observe that MGT's cross-attention mechanisms naturally provide informative cues for adaptive localization of edit-relevant regions, albeit with insufficient prominence and limited focus clarity. Focusing on this drawback, we propose a *multi-layer attention consolidation* that enhances attention weights, rendering target editing regions more distinctive and thereby achieving *capability* ❶ as demonstrated in Figure 3. Furthermore, we introduce *region-hold sampling* that realizes *capability* ❷ by constraining token modifications in low-attention areas, effectively enabling the model to concentrate on semantically meaningful regions, thus explicitly resolving the problem.

Given the scarcity of high-resolution image editing datasets, we constructed CrispEdit-2M across 7 distinct categories using open-source models with rigorous filtering procedures to ensure quality. Using 5M collected samples, we trained EditMGT based on Meissonic (Bai et al., 2024b), leveraging attention injection mechanisms that incorporate the input image as additional conditioning to supervise generation without introducing additional parameters.

We demonstrate the effectiveness of EditMGT through comprehensive experiments on standard benchmarks encompassing three pixel-level similarity metrics and one GPT-based semantic evaluation. Despite our model's compact size of only 960M parameters – $2\times$ to $8\times$ smaller than existing baselines – we achieve state-of-the-art performance on image similarity metrics across Emu Edit and MagicBrush benchmarks, optimal performance across all AnyBench task categories with substantial improvements of 3.6% for style change and 1.7% for implicit instruction tasks, and nearly competitive results comparable to the 12B FluxKontext.dev model on GEdit-EN-full with a remarkable 17.6% improvement in style transfer. Additionally, our approach surpasses VAREdit-8B, GoT-6B, and OmniGen2-7B, demonstrating superior overall performance. Furthermore, as shown in Figure 1(b), EditMGT achieves $6\times$ faster editing speed compared to models with similar performance on $1024 \times 1024$ images (requiring only 2 seconds per edit), while maintaining a memory footprint of merely 13.8 GB, thereby providing a new foundation for the image creation community.

In summary, this paper makes three contributions to the image editing community:

- We introduce **EDITMGT**, the first MGT-based image editing model that fundamentally addresses the spurious edit leakage problem in DMs by leveraging MGT's token flipping nature to explicitly preserve edit-irrelevant regions.
- We propose multi-layer attention consolidation with region-hold sampling to achieve adaptive localization of edit-relevant regions, solving the challenge of determining where edits should be applied without requiring manually predefined masks.
- We construct CrispEdit-2M, a high-resolution ($\geq$1024) image editing dataset spanning 7 distinct categories with 2M rigorously filtered samples.
- Extensive experiments on four popular benchmarks validate the effectiveness of our approach, with our compact 960M model achieving $6\times$ faster editing than comparable methods.

## 2 RELATED WORK

**Masked Generative Transformer (MGT)** is an emerging architecture for efficient text-to-image generation (Chang et al., 2022; 2023; Patil et al., 2024). It encodes images as discrete sequences of visual tokens using a VQ-GAN encoder (Esser et al., 2021), then trains a bidirectional transformer (Devlin et al., 2019) to model natural image distributions in the discrete token sequence space. Generation is performed iteratively, where significant efficiency gains are achieved through parallel sampling (Ghazvininejad et al., 2019), generally resulting in faster inference speeds. Meissonic (Bai et al., 2024b) extended MGT to $1024 \times 1024$ resolution while matching SDXL (Podell et al., 2023) performance through multimodal attention mechanisms and improved noise scheduling. Previous applications of MGT have been primarily limited to inpainting (Ko & Kim, 2023; Kim et al., 2023) and interpolation (Ma et al., 2024). To the best of our knowledge, EditMGT represents the first MGT-based image editing framework.

**Image Editing** InstructPix2Pix (Brooks et al., 2023) established the paradigm of fine-tuning text-to-image models into editing models using instruction, source image, edited image triplets. Subsequent research has pursued two primary directions for improvement: enhancing the quality and complexity of training data (Zhang et al., 2024; Yu et al., 2025; Ge et al., 2024a; Wang et al., 2025b; Ye et al., 2025), and advancing the capabilities of the underlying generative architecture (Labs et al., 2025; Wu et al., 2025a; Team et al., 2023). While the majority of existing editing techniques primarily focus on DM-based approaches, the global denoising dynamics inherent to DMs introduce the problem of editing leakage. EditMGT represents the first MGT-based editing model and demonstrates effective mitigation of this issue. Due to space limits, the additional related work in image editing are placed in Appendix Sec. D and Table 8.

**Attention Control** Recent DiT-based diffusion models leverage attention mechanisms to capture rich semantic features for image editing. MasaCtrl (Cao et al., 2023) introduces mutual self-attention to retrieve semantically correlated content from source images, ensuring coherent edits. Prompt-to-Prompt (Hertz et al., 2022) modulates text-image relationships via cross-attention layers, an approach widely adopted in subsequent works (Chen et al., 2024a; Yang et al., 2023; Parmar et al., 2023). DiTCtrl (Cai et al., 2025a) employs controlled attention to decouple foreground and background elements, enabling independent editing with temporal coherence in videos. To the best of our knowledge, EditMGT presents the first systematic analysis of full attention dynamics in MGT during token flipping and leverages this understanding to mitigate editing leakage.

## 3 EDITMGT: TOWARDS MGT-BASED IMAGE EDITING

In this section, we present the technical details of the proposed EditMGT. In Section 3.1, we introduce the architectural implementation of MGT-based editing, which leverages attention injection to achieve image editing without introducing additional parameters. Then, in Section 3.2, we illustrate the inference procedure. Focusing on the analysis of the attention mechanism in MGT models, we propose a multi-layer attention consolidation coupled with region-hold sampling to exploit this mechanism, ensuring the preservation of irrelevant regions during inference. Finally, in Section 3.3, we describe the training procedure of EditMGT with the proposed CrispEdit-2M dataset.

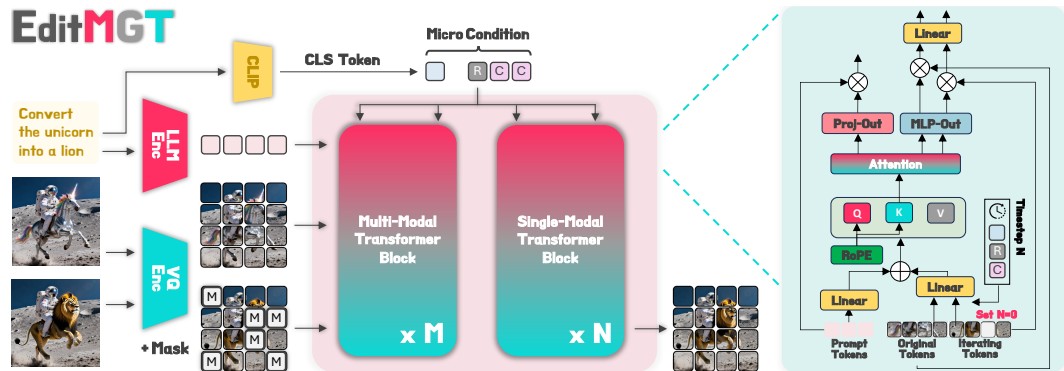

Figure 2: **Overview of EDITMGT**. Our approach supervises edited image generation through original image attention injection. The right panel illustrates token-wise interactions within the multimodal transformer block, while the single-modal block adopts an analogous architecture. Detailed descriptions of the attention injection mechanism and iterative paradigm are provided in Section 3.1.

## 3.1 ARCHITECTURE

**Preliminary.** MGT starts from a blank canvas where all visual tokens are masked. At each sampling iteration, all missing tokens are sampled in parallel, and a rejection criterion is used, where the tokens with low model likelihood are masked and will be re-predicted in the next refinement iteration. We define the image and text condition tokens as $C_I \in \mathbb{R}^{N \times d}$ and $C_T \in \mathbb{R}^{M \times d}$, where $d$ is the embedding dimension, and $N$, $M$ are their respective token counts.

For the implementation of Meissonic (Bai et al., 2024b), each transformer block first applies rotary position embedding (RoPE) (Su et al., 2024) to encode the tokens. For image tokens $C_I$, RoPE applies rotation matrices based on the token's position $(i, j)$ in the 2D grid: $C_{I i,j} \rightarrow C_{I i,j} \cdot R(i, j)$, where $R(i, j)$ denotes the rotation matrix at position $(i, j)$. Text tokens $C_T$ undergo the same transformation with their positions set to $(0, 0)$. The multi-modal attention mechanism then projects the concatenated position-encoded tokens $C = [C_I; C_T]$ into query $Q$, key $K$, and value $V$ representations. We can calculate attention weight: $\mathbf{W} = \text{softmax}(\frac{QK^\top}{\sqrt{d}})$. Then, the product of $\mathbf{W}$ and $V$ is passed through a normalization layer (Ba et al., 2016) before being propagated to the next module. $\mathbf{W}$ is endowed with rich semantic information, and we subsequently incorporate additional image conditions based on attention weights, while introducing both local and global guidance during inference.

**Image Conditional Integration.** To let the raw image supervise the image generation process, we further define image condition tokens $C_V \in \mathbb{R}^{N \times d}$, which have the same shape with $C_I$. Specifically, we let the RoPE matrices: $(i, j)_{C_V} = (i, j)_{C_I}$, which ensures spatial alignment between the original and edited images. As illustrated in the right side of Figure 2, $C_V$ shares parameters with $C_I$ and undergoes identical iterative updates, with the critical distinction that the timestep for $C_V$ remains fixed at zero throughout the process. This design choice prevents drift in $C_V$, maintaining its role as a stable conditioning signal.

During training, the model $\theta$ is optimized via minimizing the negative log-likelihood of reconstructing masked tokens conditioned on both unmasked tokens and the condition tokens on a large-scale image-text dataset $\mathcal{D}$, $\mathcal{M}$ means the masked tokens:

$$L = \mathbb{E}_{(x,t) \sim \mathcal{D}, \mathbf{m} \sim \mathcal{M}} \left[ - \sum_{i \in \mathbf{m}} \log p_\theta(v_i | v_{\neg i}, C_T; C_V) \right]. \tag{1}$$

where $v \in C_I$, $\mathbf{m} \sim \mathcal{M}$ is a binary mask applied to the tokens, selecting indices $i$ to mask, $v_{\neg i}$ refers to the unmasked tokens, and $p_\theta(v_i | v_{\neg i}, C_I; C_V)$ is the predicted probability of token $v_i$. We use cosine scheduling strategy during training, with a masking rate $r \in [0, 1]$ is sampled from a truncated $\arccos$ distribution, with the density function $p(r) = \frac{2}{\pi}(1 - r^2)^{-\frac{1}{2}}$.

To control the strength of $C_V$ during inference, following Tan et al. (2024), we introduce a bias term $\mathcal{E}$ into the attention weight as $\mathbf{W}_{new} = \mathbf{W} + \mathcal{E}$, where $\mathcal{E}$ is a bias matrix modulating the attention

Figure 3: Attention Mechanism in EditMGT. The text-to-image attention maps encode rich semantic correspondences. We enhance their clarity through stacking and filtering operations.

between concatenated tokens $[C_T; C_I; C_V]$. This process can be formulated as follows:

$$\mathcal{E} = \begin{bmatrix} \mathbf{0}_{M \times M} & \mathbf{0}_{M \times N} & \mathbf{0}_{M \times N} \\ \mathbf{0}_{N \times M} & \mathbf{0}_{N \times N} & \log(\gamma)\mathbf{1}_{N \times N} \\ \mathbf{0}_{N \times M} & \log(\gamma)\mathbf{1}_{N \times N} & \mathbf{0}_{N \times N} \end{bmatrix}. \tag{2}$$

This formulation preserves the original attention patterns within each token type while scaling attention weights between $C_I$ and $C_V$ by $\log(\gamma)$. At test time, setting $\gamma = 0$ removes the condition's influence, while $\gamma > 1$ enhances it. Through this approach, we seamlessly embed conditioning via attention mechanisms, achieving the transformation from a text-to-image model to an editing model without introducing additional parameters.

## 3.2 INFERENCE

Building upon the above architecture, we observe that the cross-attention mechanisms in EditMGT naturally provide informative cues for adaptive localization of edit-relevant regions. As illustrated in Figure 3, we investigate the cross-attention mechanism between the iterative image $C_I$ and instruction $C_T$ (due to space constraints, we omit the cross-attention visualization between the original image $C_V$ and these two modalities). Our analysis reveals that each text-to-image attention weight in the MGT model contains rich semantic information, establishing effective correspondence between textual instructions and visual regions. Remarkably, the model can predict the styling of key regions in the edited image within the initial iterations. For instance, in the example *"add a birthday hat on the dog"*, MGT directly delineates the contour of the hat shape.

**Multi-layer Attention Consolidation.** Raw attention weights from individual intermediate blocks exhibit insufficient prominence and lack clear focus, even when extracted from the most coherent layers. To address this limitation, we propose a multi-layer attention consolidation that systematically enhances attention clarity. Specifically, we aggregate attention weights from blocks 28 through 36, selected from coherent single-modality processing layers, to amplify signal strength. However, we observe that the aggregated attention weights still manifest incomplete activation regions characterized by internal discontinuities and poorly-defined boundaries, potentially leading to erroneous token classifications within object interiors. To mitigate these artifacts, we incorporate Adaptive Filtering (Diniz et al., 1997) to achieve enhanced clarity and spatial precision. Implementation details are provided in Appendix B.2.

**Region-Hold Sampling.** In the analysis of the attention mechanism, we observe that the attention weights of MGT exhibit rich semantic information, enabling a well-aligned text-to-image correspondence. During image generation, MGT progressively refines the target image through iterative token flipping. As illustrated in Figure 4, EditMGT accurately localizes the key regions for editing. Consequently, we preserve the unmodified regions by explicitly flipping the low-attention areas back to their original tokens.

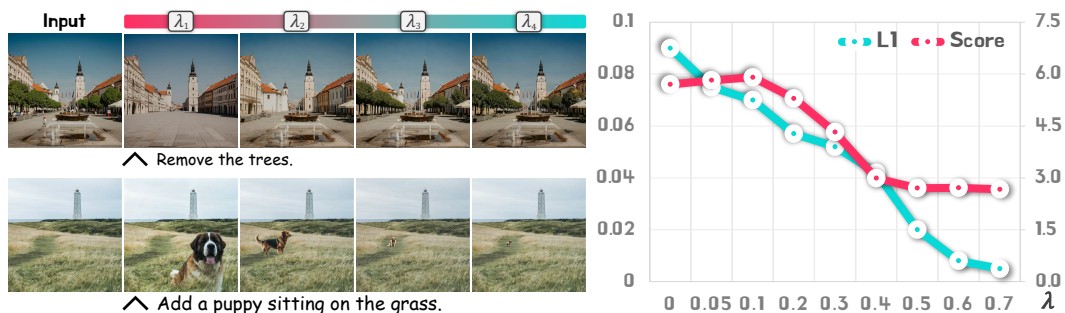

Figure 4: **Visualizations** of editing results, GEdit Bench semantic scores, and L1 distances from original images across varying threshold $\lambda$. Additional details can be seen in Appendix Sec. C.6.

We define $\mathcal{W}_v^l, \mathcal{W}_i^l \in \mathbb{R}^{M \times N}$ as the attention maps from $C_T \rightarrow C_V$ and $C_T \rightarrow C_I$ after normalization at layer $\ell$ respectively. To flexibly control the flipping frequency, we introduce a threshold $\lambda$ to determine which tokens should be restored to the original image. Specifically, we can obtain the localization map as follows:

$$s_L = \frac{1}{|\mathcal{L}||\mathcal{M}|} \sum_{\ell \in \mathcal{L}, m \in \mathcal{M}} \mathcal{W}_i^l[m, :] \in \mathbb{R}^N \,, \tag{3}$$

where $\mathcal{W}_i^l[m, :]$ denotes the $m$-th row slice of the matrix $\mathcal{W}_i^l$, $\mathcal{M}$ is the set of all row indices to be selected, and $|\mathcal{M}| \leq M$ with equality holding if and only when the entire $\mathcal{W}_i^l$ is selected. If we only use the keywords from the instruction, such as a specific object, then we can extract the corresponding portion using $\mathcal{M}$. During inference, EditMGT flips tokens with high confidence while keeping low-confidence tokens as [MASK] for subsequent refinement. With the introduced sampling method, tokens satisfying $S_L < \lambda$ are reverted to their original counterparts, thereby preserving both the sampling scheduler's integrity and consistency with the source image. Figure 4 illustrates the relationship between edited images and $\lambda$ - when $\lambda$ exceeds a certain threshold, the output becomes identical to the original image.

### 3.3 TRAINING DETAILS

Given the scarcity of high-resolution image editing datasets, we constructed CrispEdit-2M across 7 distinct categories. CrispEdit-2M comprises 2M samples with short edge $\geq 1024$ pixels generated using open-source models, employing rigorous filtering procedures to ensure data quality. Combined with an additional 2M high-resolution samples we collected, we utilized a total of 4M image editing data samples for training. The detailed data construction pipeline and comprehensive statistics are provided in Appendix Sec. A.

We implement EditMGT based on Meissonic (Bai et al., 2024b). Since Meissonic exhibits a bias toward generating cartoon-style content and employs CLIP as the text encoder, which lacks strong language understanding capabilities (Xie et al., 2024; Gong et al., 2025; Gao et al., 2025) – a critical requirement for edit models – we divide EditMGT's training into **three phases**.

**Stage ❶: Base Model with an LLM**, of which we utilize approximately 1M text-image pairs and directly employ Gemma2-2B-IT (Team et al., 2024b) as the text encoder, training for 5,000 steps.

**Stage ❷: Full-Tune Edit Model** on the complete 4M image edit dataset for 50,000 steps.

**Stage ❸: High-Quality Full-Tune** the model for 1,000 steps using the higher-quality editing data to enhance alignment between the model outputs and human preferences.

Due to space limits, more training details have been placed in Appendix Sec. C.1.

## 4 EXPERIMENTS

To validate the effectiveness of EditMGT, we conduct comprehensive evaluations in Section 4.1 on three pixel-level benchmarks (Emu Edit, MagicBrush, and AnyBench) and one GPT-based evalua-

Table 1: **Comparative results** for instructive image editing on the test sets of EMU Edit (Sheynin et al., 2024) and MagicBrush (Zhang et al., 2024). We list the task-specific models in the first block and some concurrent universal models in the second block.

| Method | EMU Edit Test Set | | | | MagicBrush Test Set | | | |
|---|---|---|---|---|---|---|---|---|
| | CLIP$_{im}$↑ | CLIP$_{out}$↑ | L1↓ | DINO↑ | CLIP$_{im}$↑ | CLIP$_{out}$↑ | L1↓ | DINO↑ |
| InstructPix2Pix (Brooks et al., 2023) | 0.834 | 0.219 | 0.121 | 0.762 | 0.837 | 0.245 | 0.093 | 0.767 |
| MagicBrush (Zhang et al., 2024) | 0.838 | 0.222 | 0.100 | 0.776 | 0.883 | 0.261 | 0.058 | 0.871 |
| PnP (Tumanyan et al., 2023) | 0.521 | 0.089 | 0.304 | 0.153 | 0.568 | 0.101 | 0.289 | 0.220 |
| Null-Text Inv. (Mokady et al., 2023) | 0.761 | 0.236 | 0.075 | 0.678 | 0.752 | 0.263 | 0.077 | 0.664 |
| UltraEdit (Zhao et al., 2024) | 0.793 | 0.283 | 0.071 | **0.844** | 0.868 | - | 0.088 | 0.792 |
| EMU Edit (Sheynin et al., 2024) | 0.859 | 0.231 | 0.094 | 0.819 | 0.897 | 0.261 | 0.052 | 0.879 |
| AnyEdit (Yu et al., 2025) | 0.872 | 0.285 | 0.070 | 0.821 | 0.898 | 0.275 | **0.051** | 0.881 |
| OmniGen (Xiao et al., 2025) | 0.836 | 0.233 | - | 0.804 | - | - | - | - |
| PixWizard (Lin et al., 2024) | 0.845 | 0.248 | **0.069** | 0.798 | 0.884 | 0.265 | 0.063 | 0.876 |
| UniReal (Chen et al., 2024b) | 0.851 | 0.285 | 0.099 | 0.790 | 0.903 | **0.308** | 0.081 | 0.837 |
| GoT-6B (Fang et al., 2025) | 0.864 | 0.276 | - | - | - | - | - | - |
| OminiGen2 (Wu et al., 2025b) | 0.876 | **0.309** | - | 0.822 | - | - | - | - |
| EditAR (Mu et al., 2025) | - | - | - | - | 0.867 | - | 0.103 | 0.804 |
| NEP (Wu et al., 2025c) | 0.871 | 0.307 | 0.078 | **0.844** | - | - | - | - |
| VAREdit-8B (Mao et al., 2025) | 0.876 | 0.280 | 0.094 | 0.825 | 0.901 | 0.287 | 0.083 | 0.844 |
| EDIT**MG**T (Ours) | **0.878** | 0.308 | 0.093 | 0.832 | **0.911** | 0.301 | 0.058 | **0.881** |

Table 2: **Comparative results** on the GEdit-EN-full benchmark (Liu et al., 2025).

| Model | BG Change | Color Alt. | Mat. Mod. | Motion | Port. | Style | Add | Remove | Replace | Text | Tone | Avg |
|---|---|---|---|---|---|---|---|---|---|---|---|---|
| AnyEdit | 4.31 | 4.25 | 2.64 | 0.67 | 1.90 | 1.95 | 3.72 | 3.75 | 3.23 | 0.77 | 4.21 | 2.85 |
| MagicBrush | 6.17 | 5.41 | 4.75 | 1.55 | 2.90 | 4.10 | 4.13 | 5.10 | 1.33 | 5.07 | 4.19 |
| InstructPix2Pix | 3.94 | 5.40 | 3.52 | 1.27 | 2.62 | 4.39 | 3.07 | 1.50 | 3.48 | 1.13 | 5.10 | 3.22 |
| OmniGen | 5.23 | 5.93 | 5.44 | 3.12 | 3.17 | 4.88 | 6.33 | 6.35 | 5.34 | 4.31 | 4.96 | 5.01 |
| OminiGen2 | 6.99 | 6.66 | 4.88 | 2.55 | 3.66 | 6.08 | 7.09 | 6.60 | 6.65 | 4.49 | 6.03 | 5.57 |
| UltraEdit (SD3) | 5.83 | 5.51 | **5.86** | 3.55 | 5.00 | 5.73 | 5.06 | 3.15 | 5.79 | 2.24 | 5.45 | 4.83 |
| GoT-6B | 4.11 | 5.75 | 3.04 | 1.71 | 2.69 | 4.72 | 5.77 | 4.59 | 5.65 | 1.16 | 4.24 | 3.95 |
| VAREdit-8B | 6.77 | 6.64 | 5.40 | 3.33 | 4.20 | 6.46 | 5.86 | **7.29** | **6.67** | 3.87 | 6.54 | 5.73 |
| FluxKontext.dev | 7.06 | 7.03 | 5.52 | **5.62** | 4.68 | 5.55 | 6.95 | 6.76 | 6.13 | **6.10** | 7.48 | **6.26** |
| EDIT**MG**T (Ours) | **7.69** | **7.71** | 5.77 | 3.84 | 5.13 | 6.53 | 6.13 | 5.24 | 5.56 | 4.53 | 6.42 | 5.87 |

tion benchmark (GEdit-EN-full). We then present qualitative comparisons in Section 4.2, followed by ablation and in-depth studies in Section 4.3.

## 4.1 MAIN RESULTS

In this section, we conduct quantitative comparisons between EditMGT and baseline methods across four benchmark datasets. Detailed information regarding the baseline methods and evaluation metrics can be found in Appendix C.4 and C.5, respectively.

**Emu Edit & MagicBrush.** As shown in Table 1, our model achieves state-of-the-art performance in image similarity as measured by CLIP$_{im}$ scores across all evaluated models, with a notable improvement of 1.1% on MagicBrush. For semantic image similarity evaluated using DINO, our approach attains second-best and state-of-the-art results on Emu Edit and MagicBrush, respectively. The instruction adherence metrics demonstrate consistently strong second-best performance, indicating that our model effectively follows editing instructions. While our L1 scores do not show significant advantages compared to other baselines, this may be attributed to the inherent diversity differences between EditMGT and the predetermined target images.

**AnyBench.** As illustrated in Figure 5(a)(b), EditMGT achieves either optimal or near-optimal performance across all tasks in the AnyBench evaluation when categorized by task type. Notably, for style change tasks, EditMGT demonstrates a substantial improvement of 3.6% over the second-best performing method. For implicit instruction tasks, EditMGT consistently achieves SOTA results, outperforming the second-best model by 1.7%, indicating our model's superior capability in handling implicit instructional guidance. Detailed scores for AnyBench are provided in Tables 6 and 7.

**GEdit-EN-full.** We further evaluate our model on the GEdit-EN-full benchmark, which employs GPT-based assessment encompassing both generation accuracy and image quality. We report the overall performance metrics in Table 2. Despite our model's compact size of only 960MB, it

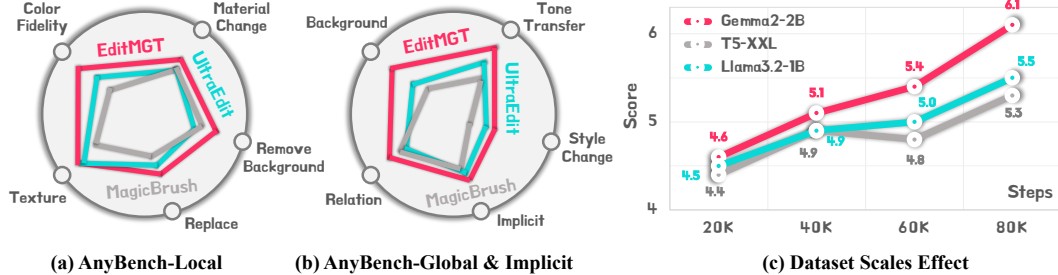

(a) AnyBench-Local     (b) AnyBench-Global & Implicit     (c) Dataset Scales Effect

Figure 5: **(a)** AnyBench (local part) Results on DINOv2 scores. **(b)** AnyBench (global part and implicit part) Results on DINOv2 scores. **(c)** Ablation study on the dataset scales effect.

achieves competitive performance comparable to the 12B FluxKontext.dev model and demonstrates superior overall performance compared to VAREdit-8B, GoT-6B, and OminiGen2 (7B). Notably, our model outperforms FluxKontext.dev on several challenging tasks including background change, color change, portrait editing, and style transfer. The performance gain is particularly pronounced in style transfer, where our method achieves a 17.6% improvement over FluxKontext.dev.

## 4.2 QUALITATIVE RESULTS

Beyond quantitative metrics for evaluating editing tasks, we conduct qualitative evaluations by comparing our approach with UltraEdit (SD3), GoT-6B, OminiGen2-7B, and VAREdit-8B to further assess the effectiveness of our method, as illustrated in Figure 6. Notably, UltraEdit (SD3) represents a diffusion-based model with parameter count comparable to EditMGT; GoT-6B and OminiGen2-7B are unified multi-modal models; while VAREdit is a VAR-based architecture. It is worth emphasizing that our model contains only 960MB parameters, whereas the compared baselines range from $2\times$ to $8\times$ larger in parameter count.

Our key observations are as follows: (i) EditMGT demonstrates superior instruction comprehension capabilities. For instance, in the case *"My photo looks a bit yellowish; please adjust the color,"* other models erroneously interpret this as a request to increase yellow tones, whereas only EditMGT correctly reduces warm tones to achieve better skin whitening and enhanced visual aesthetics. (ii) EditMGT exhibits robust object attribute understanding. In the example *"Light all the candles to enhance the candlelight,"* only EditMGT successfully illuminates all candles; for *"Add long black stockings,"* it accurately comprehends the adjective modifier "long"; and in *"Add a robot bird in the sky,"* it correctly generates a mechanical bird rather than a conventional bird as produced by other models. (iii) EditMGT effectively preserves original structural composition. In the case *"Generate a Pixar-style animation with a cheerful spring background,"* we not only successfully render the fox-like character but also maintain the original pose and positioning of the subject.

## 4.3 IN-DEPTH ANALYSIS

*(i) Data Scaling*. To evaluate the scalability of our proposed method, we conduct experiments across different training steps and report the Overall scores on GEdit-Bench as shown in Figure 5. Our results demonstrate that the model architecture maintains consistent scalability even when the text encoder is replaced, indicating robust performance across various training regimes. *(ii) Architecture Ablation*. We primarily investigate the choice of text encoder in our model architecture. Following the experimental setup outlined in Table 5, we train our model with different text encoder configurations. Our empirical analysis reveals that Gemma2-IT-2B achieves the best performance among the evaluated alternatives, establishing it as the optimal choice for our framework. *(iii) Inference Algorithm Effectiveness*. As illustrated in Figure 4, increasing values of $\lambda$ progressively reduce the extent of edited regions within the image. In the first case, fewer trees are removed from the scene until no editing occurs, while in the second case, the introduced dog becomes increasingly subtle. Correspondingly, the L1 distance to the original image decreases, whereas the semantic score exhibits an initial marginal improvement followed by a sharp deterioration.

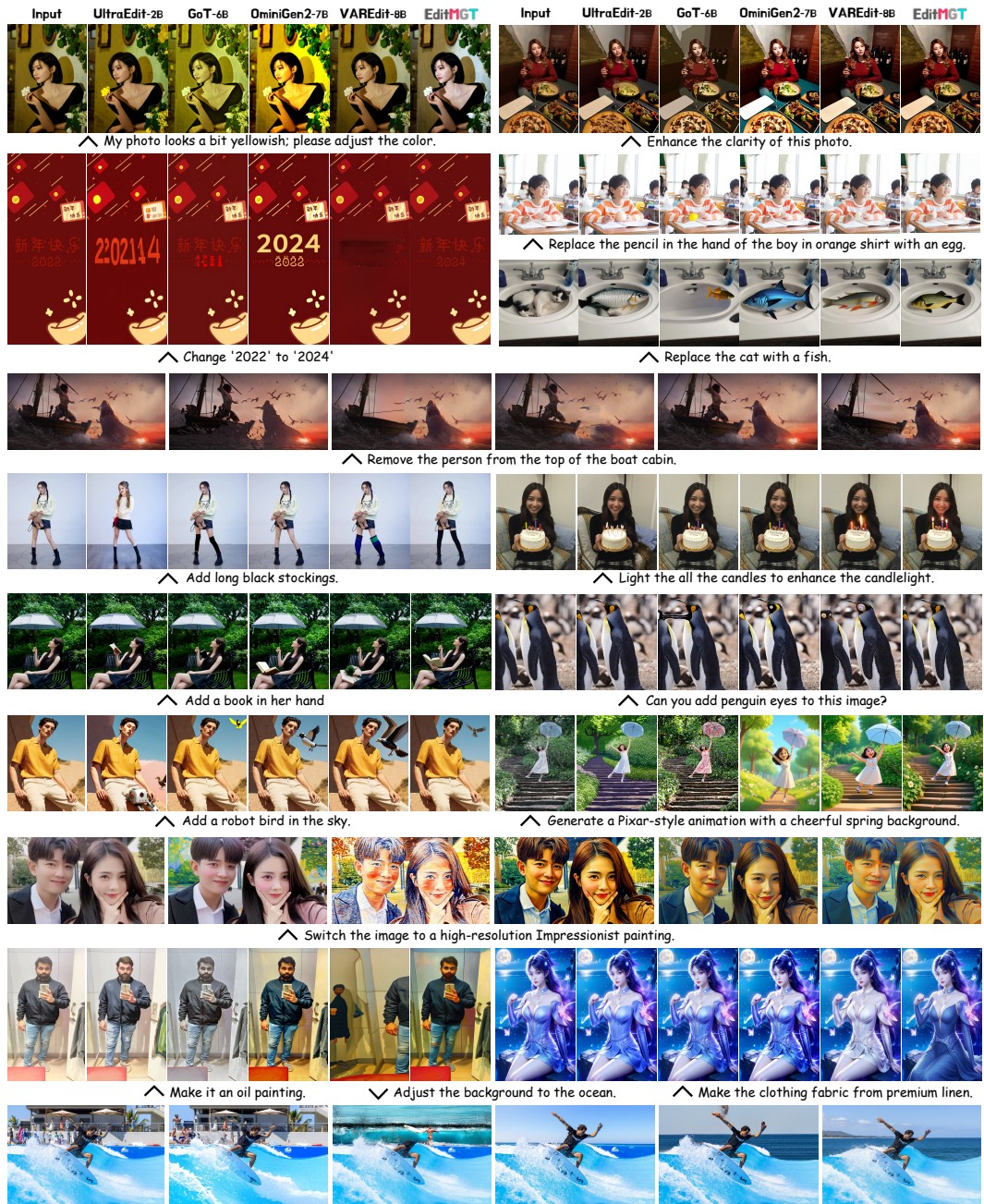

Figure 6: Qualitative comparisons between EDITMGT and other open-sourced editing models.

## 5 CONCLUSION

We presented EDITMGT, the first MGT-based image editing framework that leverages the localized decoding paradigm of masked generative transformers to address the editing leakage problem inherent in diffusion models. Through our proposed multi-layer attention consolidation and region-hold sampling, EditMGT achieves precise edit localization while explicitly preserving non-target regions. Despite using only 960M parameters, our model attains state-of-the-art image similarity performance across four benchmarks, with significant improvements of 3.6% and 17.6% on style change and style transfer tasks, respectively. Furthermore, EditMGT delivers 6× faster editing speed, demonstrating that MGTs offer a compelling alternative approach for image editing.

**Ethics Statement**: Discussion on limitations, border impacts, reproducibility, and the usage of LLMs are placed in Appendix Sec. E and Sec. F.

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

CONTENTS

# A  DATASET ANALYSIS

## A.1  CRISPEDIT-2M COLLECTION PROCESS

In this section, we provide a comprehensive description of the data collection methodology for CrispEdit-2M. As illustrated in Figure 7, the construction of CrispEdit-2M encompasses 4 stages.

**Image Curation.** Prior work has shown that high-quality seed images enhance the diversity and effectiveness of image editing tasks Ge et al. (2024a); Zhao et al. (2024); Chow et al. (2024). We curate high-quality images from three sources: LAION-Aesthetics (Schuhmann et al., 2022), Unsplash Lite datasets[1], and JourneyDB (FLUX re-generated version) (Pan et al., 2023). Through systematic filtering based on the following criteria, we obtain approximately 5.5M samples. First, we retain only images with aesthetic scores above 4.5 to ensure high visual quality. We then filter images by resolution, keeping those with short-side dimensions exceeding 1024 pixels, and apply proportional scaling to resize the shorter dimension to exactly 1024 pixels. Subsequently, we employ Qwen3 Yang et al. (2025a) to evaluate image suitability for editing data generation based on their captions, effectively filtering out simple patterns, monotonous single-scene compositions, and images containing watermarks, text overlays, stickers, or logo elements. Additionally, we incorporate approximately 0.5M images with corresponding instructions from seven categories within the ImgEdit (Ye et al., 2025) dataset – style transfer, replace, alter, remove, background, add, and motion change – to augment our curation pipeline.

**Customized Instruction Generation.** To enhance data quality, we need to improve the diversity and correctness of instructions during the data annotation process. We experimented with zero-shot instruction annotation using VLMs (Chow et al., 2025a; Xu et al., 2025a), but the results were suboptimal. When in-context examples contain images, they may introduce interference for the target image to be annotated. Conversely, when examples lack visual content, the model may fail to generate appropriate instructions that satisfy the specific task type definitions. Fine-tuning VLMs for instruction annotation presents additional challenges, as the model may struggle to determine whether an image is suitable for a particular type of editing task. This approach is particularly susceptible to hallucination artifacts – for instance, when an image contains no human subjects, a fine-tuned VLM may erroneously generate instructions for action modifications, resulting in incorrect annotation (Chow et al., 2025b; Ge et al., 2024b). To address these challenges, we propose a systematic two-stage framework for generating high-quality instruction-following data. In the first stage, we employ Qwen2.5-VL (Team, 2024) to produce detailed image captions that explicitly delineate background elements, foreground objects, and their semantic attributes. The second stage leverages GPT-4o (Achiam et al., 2023) to systematically transform these descriptive captions into actionable editing instructions across multiple modalities. To ensure both diversity and consistency in instruction generation, we introduce a constrained generation paradigm that combines type-specific constraints with contextual exemplars. This approach enables the development of specialized agents for distinct editing categories, each optimized through carefully curated in-context examples. We further implement an iterative self-refinement mechanism where newly generated instruction-caption pairs are incorporated as exemplars for subsequent generations, creating a bootstrapping process that progressively enhances instruction complexity and linguistic diversity while maintaining semantic coherence (Yu et al., 2025).

**Specific Edit Pipeline.** Previous methods typically employ complex pipelines for edit data collection, with each specific editing category requiring a dedicated pipeline (Yu et al., 2025; Ye et al., 2025; Liu et al., 2025). For instance, AnyEdit (Yu et al., 2025) utilizes a two-stage pipeline to extract segmentation masks for target objects specified in editing instructions. In the first stage, it leverages GroundingDINO Liu et al. (2023c) for object localization, followed by the Segment Anything Model (SAM) Kirillov et al. (2023) for precise mask generation. Subsequently, it employs SD-Inpaint Rombach et al. (2022b) to synthesize the target image, conditioned on both the original image and the extracted segmentation mask. However, with the advancement of image editing techniques and the deployment of commercial-grade editing models, we observe that many current open-source models have achieved remarkable performance. Therefore, our data collection pipeline primarily leverages FLUX.1 Kontext (Labs et al., 2025) and Step1X-Edit v1.2 (Liu et al., 2025), subsequently employing VLMs to select the superior result as our annotation. This approach not only enhances data quality but also enriches the diversity of our dataset.

---

[1]https://github.com/unsplash/datasets

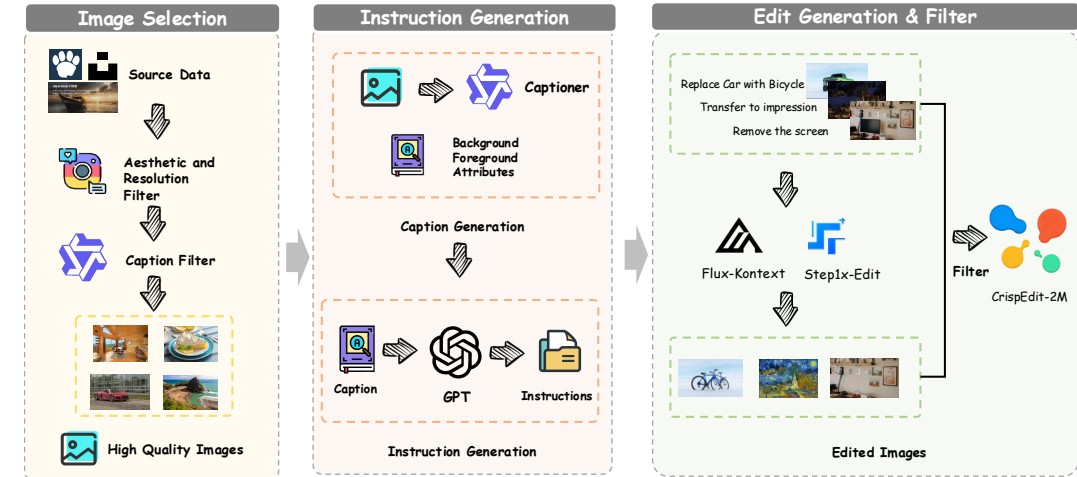

Figure 7: Overview for the CrispEdit-2M dataset collection pipeline.

**Data Quality Assurance.** We establish a comprehensive two-stage filtering framework to ensure high-quality training data throughout the annotation pipeline:

*(i) Pre-processing Instruction Validation.* LLM-generated editing instructions often contain semantic inconsistencies that compromise editing quality. Specifically, we identify two primary failure modes: (1) instructions that inadvertently modify irrelevant visual attributes (*e.g.*, altering object appearance when targeting color changes), and (2) logically inconsistent directives (*e.g.*, requesting action modifications for inherently static objects).

*(ii) Post-processing Quality Verification.* First, we leverage established CLIP-based alignment metrics Sheynin et al. (2024); Zhao et al. (2024) to quantify semantic correspondence between edited images $I_e$ and target descriptions $T_e$, ensuring faithful adherence to editing specifications within designated regions. Second, we compute CLIP-based visual similarity between source images $I_o$ and their edited counterparts $I_e$ to verify preservation of non-target content, addressing the observed tendency of FLUX.1 to generate degenerate or empty outputs under certain conditions.

## A.2 CRISPEDIT-2M STATISTICS

In this chapter, we present a coarse-grained analysis of CrispEdit-2M through resolution interval distribution plots and pie charts illustrating seven editing categories. To optimize storage efficiency, as detailed in Appendix A, we rescale the shorter dimension of our images to 1024 pixels, resulting in proportional downscaling of the entire image. Consequently, Figure 8(a) displays the distribution of the longer dimension sizes, revealing that our images are predominantly concentrated within the [1280, 1665) pixel range, thereby demonstrating the high-resolution nature of CrispEdit-2M. Concurrently, our dataset encompasses seven distinct categories, with the distribution illustrated in the pie chart presented in Figure 8(b). These categories comprise: *add* ($\approx$ 300k), *replace* ($\approx$ 300k), *remove* ($\approx$ 300k), *color alteration* ($\approx$ 500k), *background change* ($\approx$ 200k), *style transformation* ($\approx$ 400k), and *motion modification* ($\approx$ 34k).

## A.3 EDITING DATASET USAGE DETAILS

We list our used data mixture in Table 3 and we will introduce these datasets one by one:

**InstructPix2Pix (Brooks et al., 2023)** is the first publicly available editing dataset with images at a resolution of 512×512. The method employs a fine-tuned GPT-3 model to generate both editing instructions and corresponding captions for the modified images. Subsequently, pairs of images are synthesized from these caption pairs using StableDiffusion (Rombach et al., 2022a) in conjunction with Prompt-to-Prompt (Hertz et al., 2022).

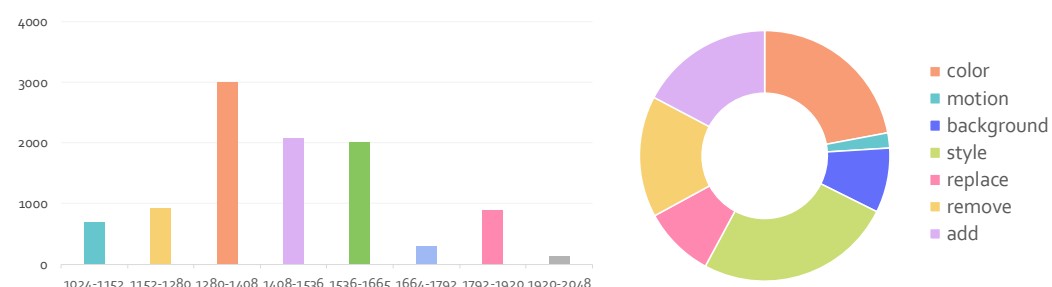

Figure 8: (a) Resolution interval distribution of CrispEdit-2M. (b) Pie chart of data types in the CrispEdit-2M dataset.

**Human-Edit** (Bai et al., 2024a) comprises 6,000 image-edit pairs annotated by human annotators using DALL·E 2. While the input images vary in size, all output images are consistently resized to a fixed resolution of $1024 \times 1024$ pixels.

**Super-Edit** (Li et al., 2025) enhances the effectiveness of supervision signals by employing vision-language models (*e.g.*, GPT-4o) to refine editing instructions, ensuring better alignment between source and edited images. Additionally, Super-Edit constructs contrastive supervision signals to further optimize the editing model. Experimental results demonstrate that Super-Edit achieves significant improvements across multiple benchmarks, outperforming existing image editing methods. The framework's key advantage lies in its ability to deliver superior editing performance without requiring additional models or pretraining tasks. Both input and output images maintain a consistent resolution of $5125 \times 512$ pixels.

**EditWorld** (Yang et al., 2024) is a benchmark dataset designed for instruction-guided image editing tasks. The dataset construction process comprises two primary pipelines: (1) text-to-image generation and (2) video frame extraction. The text-to-image generation pipeline employs GPT-3.5 and SDXL to synthesize image-edit pairs, while the video frame extraction pipeline derives image pairs from video data and utilizes video-language models Video-LLaVA (Lin et al., 2023) to generate corresponding editing instructions.

**HQ-Edit** (Hui et al., 2024) contains approximately 200,000 editing instances generated through a scalable data collection pipeline. However, it lacks fine-grained details and realism due to its diptych generation though it exploits GPT-4V (Achiam et al., 2023) and DALL-E (Ramesh et al., 2021) to enhance descriptions.

**PromptFix** (Yu et al., 2024) contains approximately 1,013,320 triplets spanning seven distinct image processing tasks: Object removal, Image dehazing, Colorization, Image deblurring, Low-light enhancement, Snow removal, Watermark removal. Each triplet consists of: (1) an input image, (2) its processed counterpart, (3) an instructional text, and (4) an auxiliary prompt generated by the InternVL2 model (except for the object removal task).

**ImgEdit** (Ye et al., 2025) comprises 1.2 million carefully curated image-edit pairs spanning 13 distinct editing categories, including both single-round operations (*e.g.*, addition, removal, replacement, modification, background alteration, and blending) and multi-round tasks (*e.g.*, content memorization, content understanding, and version backtracking). This dataset is characterized by its high image resolution, detailed editing instructions, and precise editing outcomes. The construction pipeline involves four key phases: (1) *Data Preparation* – selecting high-quality images from LAION-Aesthetics (Schuhmann et al., 2022) and generating concise captions using GPT-4o; (2) *Instruction Generation* – creating editing instructions via GPT-4o based on image captions, edit types, and target objects; (3) *Edit Generation* – producing edited images using state-of-the-art generative models (FLUX and SDXL); and (4) *Post-processing* – employing GPT-4o for quality assessment and subsequent filtering of the edited results.

**ByteMorph-6M** (Chang et al., 2025)   is a large-scale dataset comprising 6.4 million image-edit pairs spanning 5 distinct motion categories: (1) *Camera Zoom*, involving changes in camera focal length while capturing the scene; (2) *Camera Move*, entailing camera positional shifts; (3) *Object Motion*, where objects within the image undergo movement; (4) *Human Motion*, depicting articulated human motions; and (5) *Interaction*, capturing dynamic engagements between humans and/or objects. The dataset is synthetically generated using the video-based diffusion model Seaweed (Seaweed et al., 2025), ensuring natural and temporally consistent edits. Additionally, ByteMorph-6M provides detailed edit instructions and per-frame textual descriptions to facilitate model training and enhance understanding of image-editing tasks.

**OmniEdit** (Wei et al., 2024)   comprises 1.2 million samples generated through multiple expert models, constructed via a three-stage pipeline: (1) *Data Collection* – high-resolution images with diverse aspect ratios are sampled from the LAION-5B (Schuhmann et al., 2022) and OpenImageV6 (Krasin et al., 2017) databases; (2) *Expert Model Processing* – seven specialized models (*e.g.*, object replacement, removal, and addition) generate edit pairs, with each model dedicated to specific editing tasks; and (3) *Importance Sampling* – a VLM (GPT-4o and InternVL2) scores and filters the generated pairs, retaining only high-quality samples.

**GoT** (Fang et al., 2025)   consists of three distinct components: (1) *Laion-Aesthetics-High-Resolution-GoT* containing 3.77 million high-quality images filtered from Laion-Aesthetics (minimum 512-pixel resolution), annotated with prompts (mean length: 110.81 characters) and Graph-of-Thought (GoT) descriptions (mean length: 811.56 characters) generated by Qwen2-VL, averaging 3.78 bounding boxes per image; (2) *JourneyDB-GoT* comprising 4.09 million high-quality AI-generated images with Qwen2-VL-generated prompts (mean: 149.78 characters) and GoT descriptions (mean: 906.01 characters), featuring 4.09 bounding boxes per image on average; and (3) *OmniEdit-GoT* with 736K high-quality image editing samples from OmniEdit, covering diverse operations including object addition/removal/swapping, attribute modification, and style transfer.

**SEED-Data-Edit** (Ge et al., 2024a)   is a hybrid dataset for instruction-guided image editing comprises a total of 3.7 million image-editing pairs, consisting of three distinct components: (1) large-scale, high-quality editing data generated by automated pipelines (3.5M pairs), (2) real-world scenario data collected from the internet (52K pairs), and (3) high-precision, multi-turn human-annotated editing data (95K pairs, including 21K multi-turn sequences with up to 5 rounds).

**Subject-200k** (Tan et al., 2024)   is specifically designed for subject-driven image generation tasks, the Subjects200K dataset comprises over 200,000 high-quality images generated through a carefully designed pipeline to ensure subject consistency across diverse scenes. The dataset is divided into two splits: *Split-1* contains paired images of objects in different scenes, while *Split-2* pairs each object's scene images with their corresponding studio photographs. Through rigorous quality control, the dataset maintains high visual fidelity and subject consistency, providing researchers with rich training signals for learning robust subject-driven control.

**UltraEdit** (Zhao et al., 2024)   constitutes a large-scale, high-quality dataset specifically designed for instruction-based image editing tasks. The source images are collected from multiple public datasets including MS COCO (Chen et al., 2015), Flickr (Plummer et al., 2015), NoCaps (Agrawal et al., 2019), VizWiz Caption (Gurari et al., 2020), TextCaps (Sidorov et al., 2020), and Localized Narratives (Pont-Tuset et al., 2020), which provide diverse images paired with high-quality captions. The dataset creation process involves three key stages: (1) collecting high-quality image-caption pairs from various public datasets; (2) generating diverse editing instructions and corresponding target captions using LLMs combined with human annotation; and (3) producing image editing samples using real images as anchors to generate both free-form and region-specific editing samples. With approximately 4.1 million image editing samples, including around 750,000 unique editing instructions, UltraEdit covers more than nine distinct editing types such as addition, color alteration, global/local modification, transformation, replacement, and style transfer.

**HIVE** (Zhang et al., 2023b)   was constructed through a multi-stage process: initially, 1,000 images with corresponding captions were collected, and three annotators were tasked with composing three instructions and edited captions for each input caption, yielding 9,000 prompt triplets (input

Table 3: **Statistics of Existing Edit Datasets** with annotation sizes used in our study. The ✗ symbol indicates datasets excluded from our experiments. Resolution values represent the smaller dimension between input and output images. Reported sizes correspond to either training sets or complete datasets, as specified.

| Dataset | Resolution | Num (k) | Sample Num (k) | Sample Ratio (%) |
|---|---|---|---|---|
| InstructPix2Pix (Brooks et al., 2023) | 512 | 450 | ✗ | - |
| MagicBrush (Zhang et al., 2024) | 512+ | 10 | 10 | 100.0 |
| Human-Edit (Bai et al., 2024a) | 1024 | 6 | 5 | 86.7 |
| Super-Edit (Li et al., 2025) | 512 | 40 | ✗ | - |
| EditWorld (Yang et al., 2024) | 512 | 8 | ✗ | - |
| HQ-Edit (Hui et al., 2024) | 900 | 190 | ✗ | - |
| PromptFix (Yu et al., 2024) | 512+ | 1,200 | ✗ | - |
| ImgEdit (Ye et al., 2025) | 1024 | 1,000 | 100 | 10.0 |
| ByteMorph-6M (Chang et al., 2025) | 512 | 6,000 | 100 | 1.7 |
| OmniEdit (Wei et al., 2024) | 612+ | 1,200 | 900 | 75.0 |
| UltraEdit (Zhao et al., 2024) | 512 | 41,000 | ✗ | - |
| SEED-Data-Edit (Ge et al., 2024a) | 256+ | 3,700 | ✗ | - |
| Subject-200k (Tan et al., 2024) | 512 | 200 | ✗ | - |
| HIVE (Zhang et al., 2023b) | 512 | 1,100 | ✗ | - |
| AnyEdit (Yu et al., 2025) | 512+ | 2,500 | 250 | 10.0 |
| NHR-Edit (Kuprashevich et al., 2025) | 640+ | 358 | 200 | 55.9 |
| GPT-Image-Edit (Wang et al., 2025b) | 612+ | 1,500 | 500 | 33.3 |
| **CrispEdit-2M (Ours)** | 1024+ | 2,000 | 2,000 | 100.0 |
| **Total** | | | 4,065,000 | |

caption, instruction, and edited caption). These data were used to fine-tune GPT-3 for generating additional instructions and edited captions. Subsequently, BLIP was employed to generate more diverse image captions, while the Prompt-to-Prompt (Hertz et al., 2022) method based on Stable Diffusion was utilized to create paired images. The authors further developed a cycle-consistency enhancement approach through edit instruction inversion to generate supplementary data. Ultimately, the pipeline produced a total of 1.45 million training image pairs with their corresponding instructions.

**MagicBrush (Zhang et al., 2024)**   is the first large-scale, manually-annotated instruction-guided image editing dataset covering diverse scenarios single-turn, multi-turn, mask-provided, and mask-free editing. MagicBrush hires crowd workers to annotate images from the MSCOCO (Lin et al., 2014) dataset manually but only includes 10K editing pairs due to expensive labor expenses

**AnyEdit (Yu et al., 2025)**   is a large-scale dataset comprising 2.5 million high-quality image-edit pairs spanning 25 distinct editing types, which are systematically categorized into 5 primary classes: local edits, global edits, camera motion edits, implicit edits, and visual effects. The dataset ensures exceptional data quality through an adaptive editing pipeline and rigorous filtering strategies, thereby providing abundant training data for instruction-driven image editing tasks. We randomly selected 10% of the data for use during training.

## B   ATTENTION VISUALIZATION

In this chapter, we visualize three types of image-related attention maps (Fang et al., 2024) across different time steps and modules of EditMGT (total inference step is set to 32 in the Section). The MGT model has demonstrated its ability to control image generation through the attention mechanism in transformer blocks (Bashkirova et al., 2023; Esser et al., 2024; Wang et al., 2024c). However, the role of attention in editing models remains poorly understood. To bridge this gap, we further analyzed and visualized the attention maps in EditMGT in the preceding section, as illustrated in the figure below.

Since EditMGT integrates information from the pre-edited image during the attention phase and participates in the iterative generation process, we observe that the attention mechanism continues to operate on the edited (i.e., generated) image rather than the original input. Therefore, in the

following visualizations, the term image refers to the in-progress generated image, not the pre-edited one. Due to space constraints, we present only one case.

## B.1 ATTENTION WEIGHT MAP VISUALIZATION

Figure 9, Figure 10, Figure 11, and Figure 12 illustrate the attention maps generated for the editing instruction ``Put on a hat.'' in step 10. These visualizations highlight three distinct types of attention mechanisms: (1) *text-to-image* (where text tokens serve as queries and image features as keys), (2) *image-to-text*, and (3) *image-to-image*. Based on the query-key relationships within the attention maps, the transformer blocks' attention patterns can be categorized into four components. Notably, the *text-to-text* attention is omitted from our analysis due to its lack of semantic relevance in this context.

Upon examining the printed attention maps, we observe that the initial blocks in the double block structure exhibit a lack of meaningful attention information. Beginning around double block 10, some faint foreground representations emerge, though they remain indistinct. In contrast, the single block structure demonstrates more pronounced attention patterns, with several blocks clearly delineating the position of the hat – particularly in the text-to-image module.

Furthermore, we stack the attention maps from different layers. The stacked results for all 42 blocks, 14 double blocks, and 28 single blocks are illustrated in Figures 13, 14, and 15, respectively. By examining the printed attention maps, we observe that the attention in the double block is relatively dispersed. In contrast, the attention map in the single block demonstrates a more focused pattern in text-to-image tasks, accurately localizing the position where the hat should be added. Additionally, in image-to-image tasks, the single block's attention map effectively outlines the foreground and partially captures the approximate shapes of background objects. Regarding the denoising steps, as the step count increases, the foreground shapes outlined in the single block progressively align with the final generated image's foreground (while also resembling the structure of the original, unedited image).

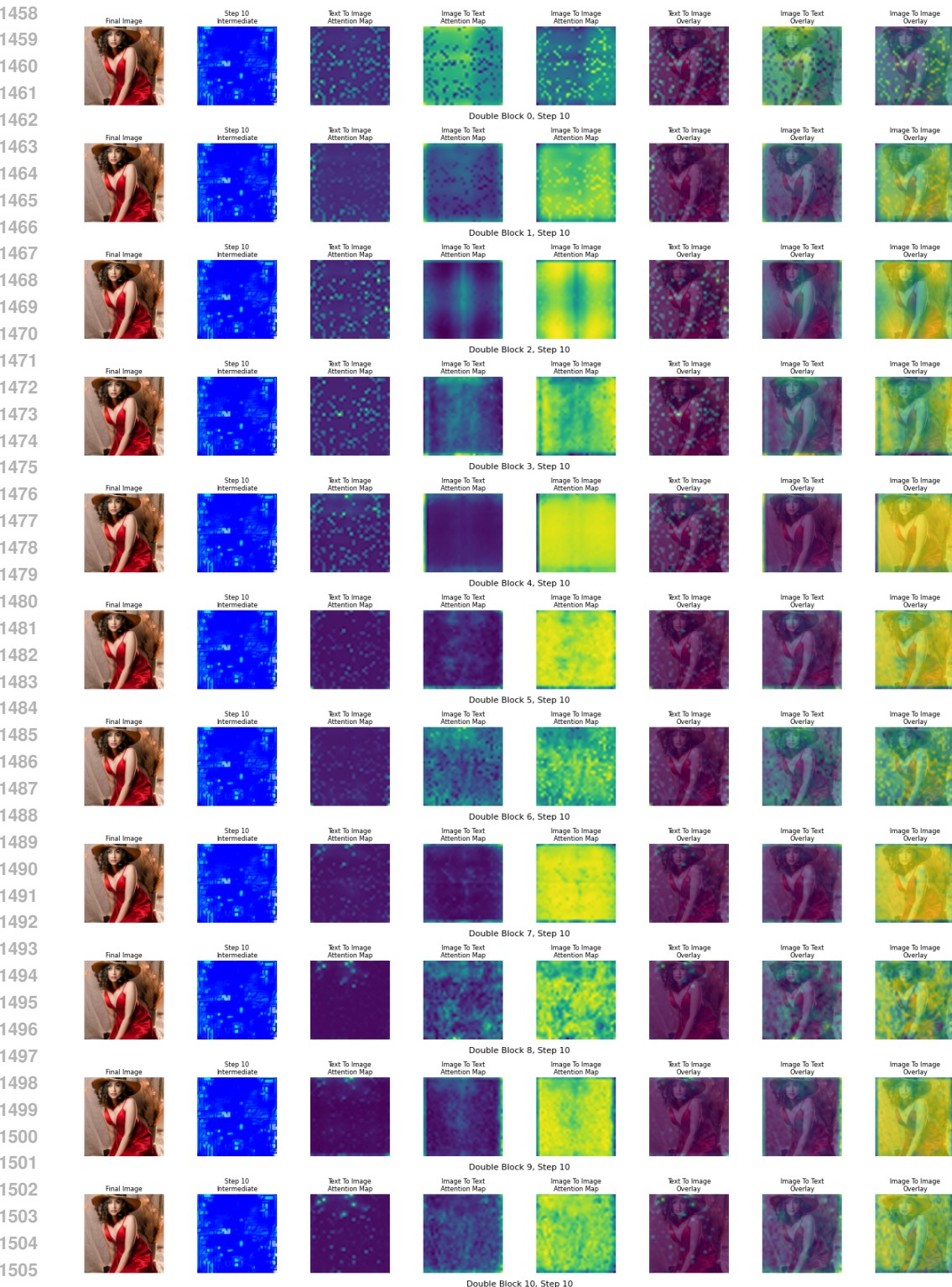

Figure 9: Attention Map Visualization for EditMGT (Transformer Block 0-10, Step 10).

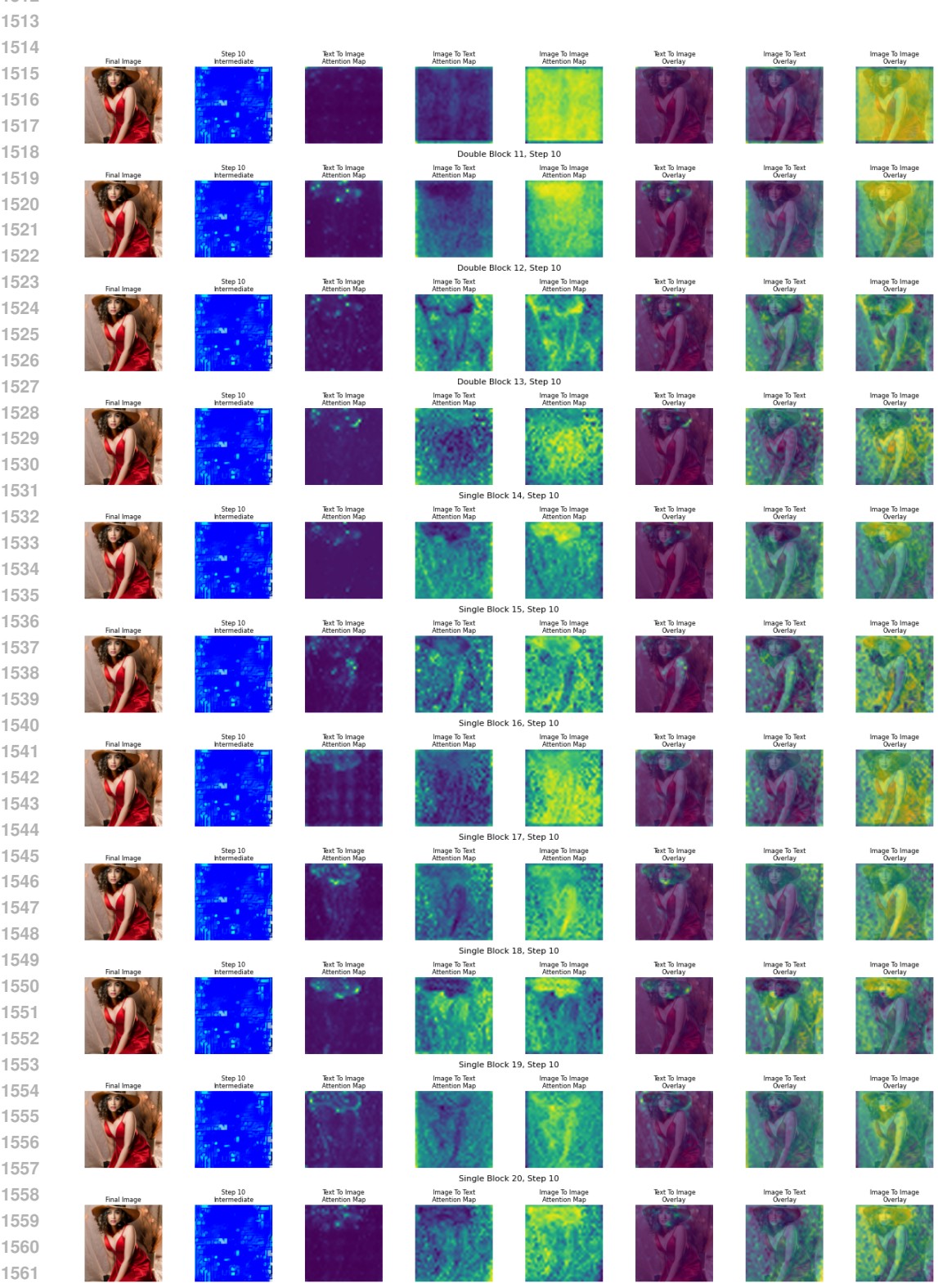

Figure 10: Attention Map Visualization for EditMGT (Transformer Block 11-21, Step 10).

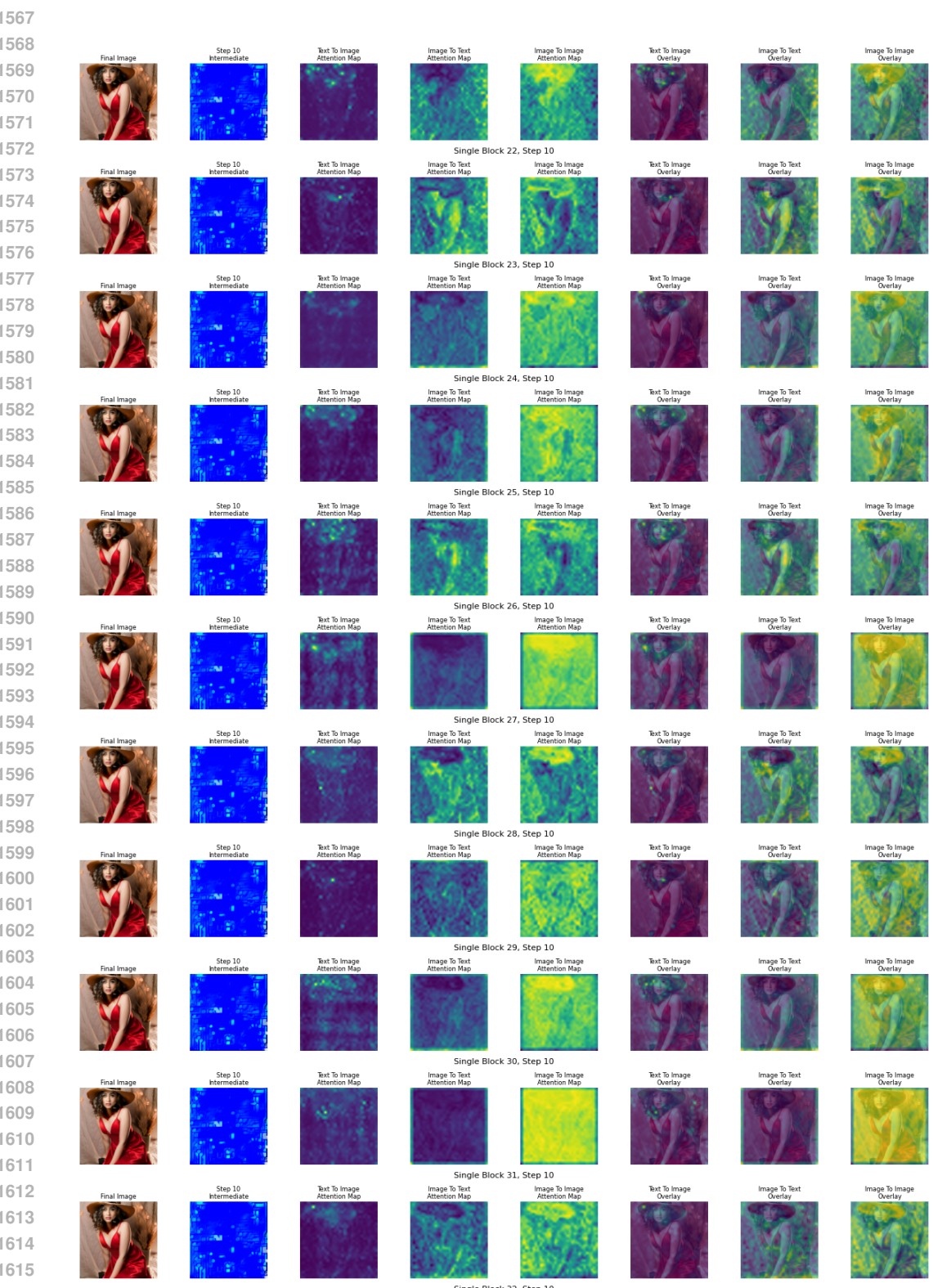

Figure 11: Attention Map Visualization for EditMGT (Transformer Block 22-32, Step 10).

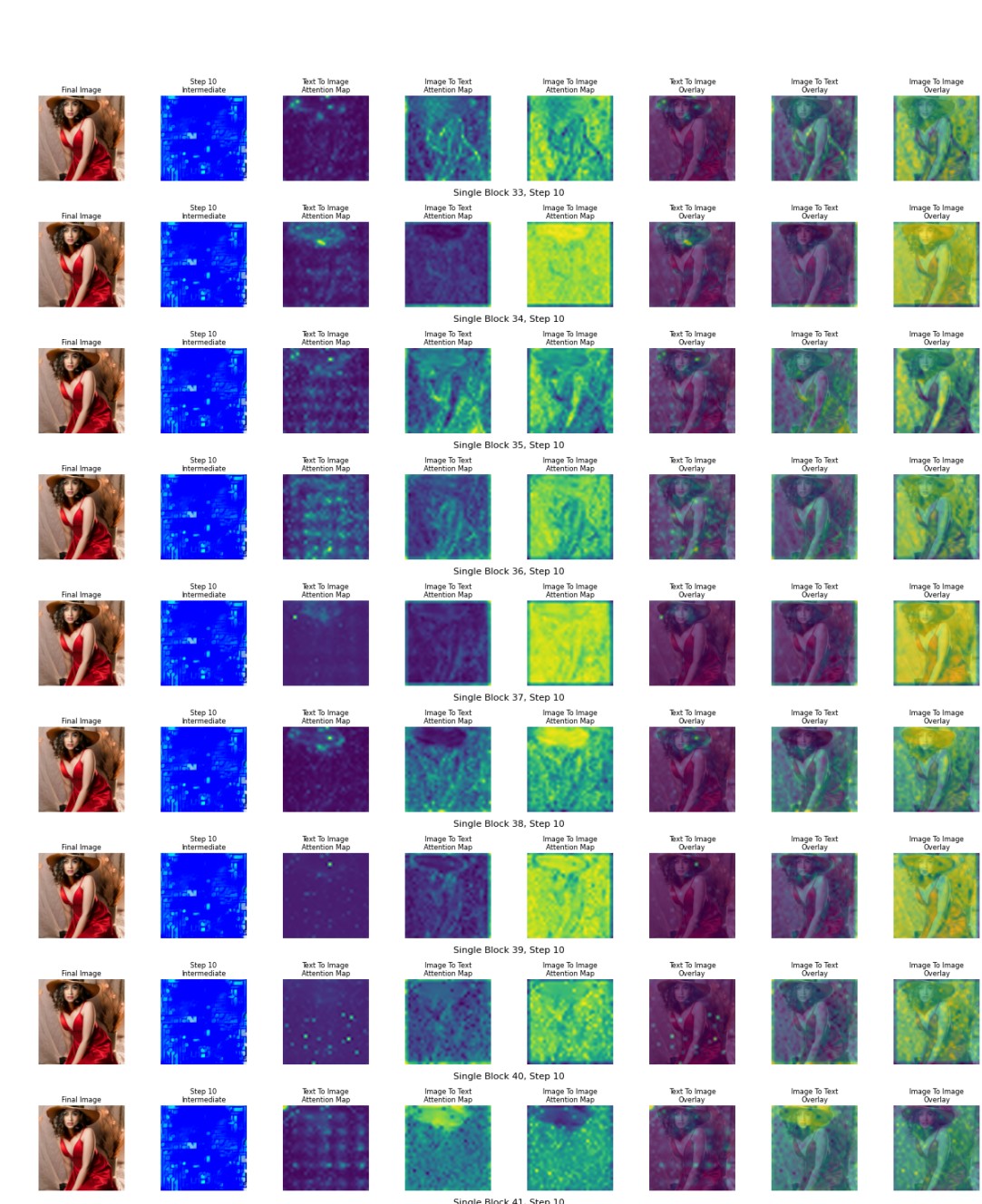

Figure 12: Attention Map Visualization for EditMGT (Transformer Block 33-41, Step 10).

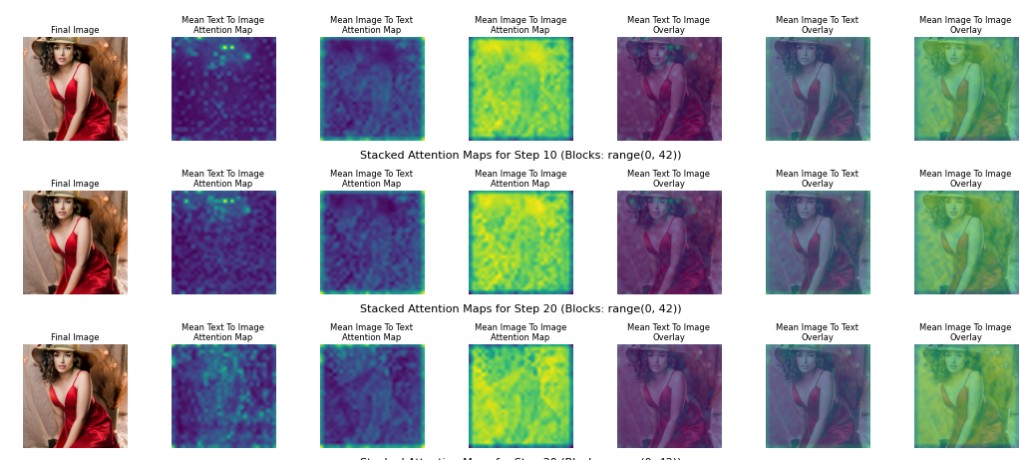

Figure 13: Attention Map Visualization for EditMGT. The attention map is stacked by all the transformer blocks (14 double blocks and 28 single blocks).

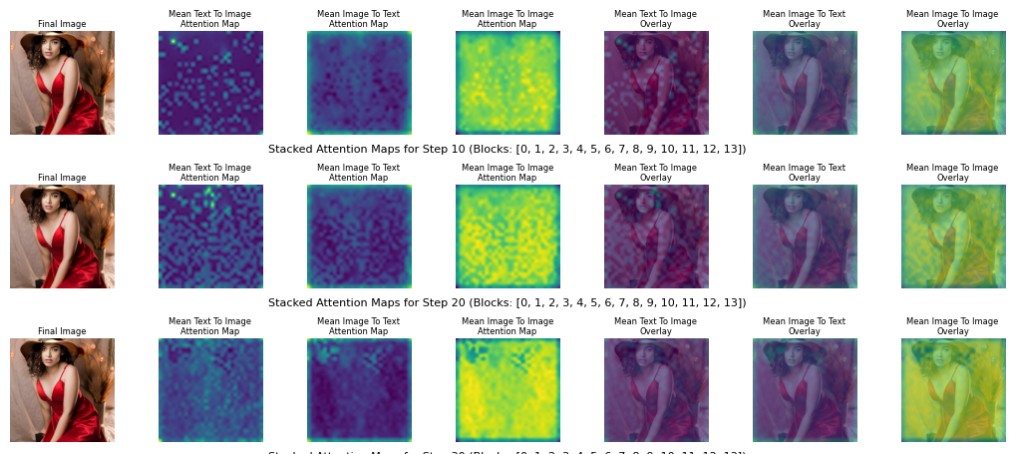

Figure 14: Attention Map Visualization for EditMGT. The attention map is stacked by all the double transformer blocks (14 blocks).

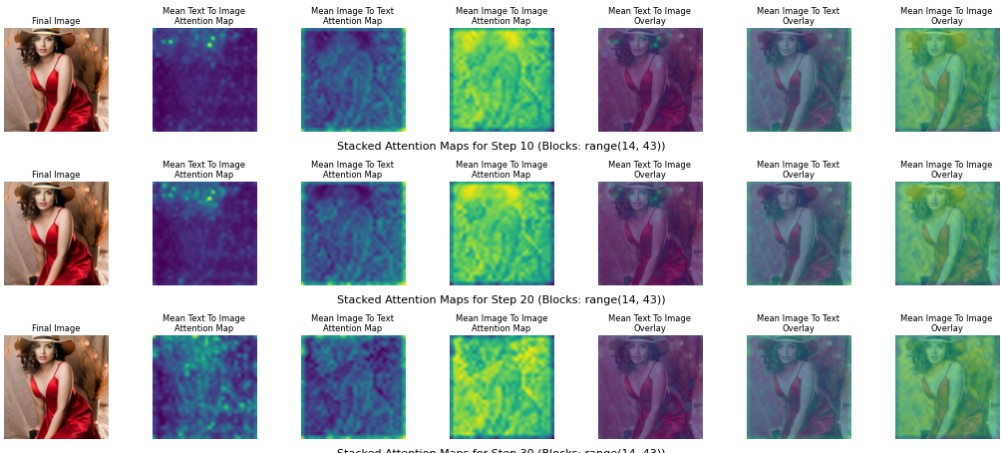

Figure 15: Attention Map Visualization for EditMGT. The attention map is stacked by all the single transformer blocks (28 blocks).

### B.2 SMOOTHENED ATTENTION WEIGHT MAP

As mentioned in Section 3, to enhance the spatial coherence of local attention scores and create more connected high-value regions, we employ four distinct filtering-based smoothing techniques (Figure 16) and four distinct interpolation-based smoothing techniques (Figure 17). These methods transform discrete token-level scores into spatially continuous representations, effectively bridging isolated high-attention areas.

Through visual analysis of the results, we observe distinct characteristics between the two smoothing paradigms. The filtering-based methods demonstrate enhanced contrast between high and low-value regions, producing steeper gradients in the attention distribution. When appropriate filtering thresholds are applied, object contours become distinctly visible, particularly with the adaptive method, as illustrated in the first column of Figure 16. In contrast, interpolation-based methods preserve value distributions more similar to the original attention maps while subtly enhancing the magnitude of neighboring values around local maxima, resulting in smoother spatial transitions with minimal alteration to the underlying attention structure.

**Filtering Methods**   *Gaussian Filtering:* Gaussian smoothing applies a Gaussian kernel to convolve with the attention map, producing isotropic smoothing that preserves the overall structure while reducing high-frequency noise:

$$G(x, y) = \frac{1}{2\pi\sigma^2} \exp\left(-\frac{x^2 + y^2}{2\sigma^2}\right) \tag{4}$$

$$I_{\text{smooth}}(x, y) = I(x, y) * G(x, y) \tag{5}$$

where $\sigma = \text{strength} \times 2.0$ controls the smoothing extent. This method provides uniform smoothing across the entire attention map, effectively connecting nearby high-attention regions.

*Bilateral Filtering:* Bilateral filtering preserves edges while smoothing homogeneous regions by considering both spatial proximity and intensity similarity:

$$I_{\text{smooth}}(x, y) = \frac{1}{W} \sum_{i,j} I(i, j) \cdot w_s(x, y, i, j) \cdot w_r(I(x, y), I(i, j)) \tag{6}$$

where $w_s$ is the spatial weight, $w_r$ is the range weight, and $W$ is the normalization factor:

$$w_s(x, y, i, j) = \exp\left(-\frac{(x - i)^2 + (y - j)^2}{2\sigma_s^2}\right) \tag{7}$$

$$w_r(I_1, I_2) = \exp\left(-\frac{(I_1 - I_2)^2}{2\sigma_r^2}\right) \tag{8}$$

This method excels at maintaining sharp boundaries between distinct attention regions while smoothing within homogeneous areas.

*Morphological Filtering:* Morphological operations use structural elements to modify the geometric structure of attention maps. We employ a combination of opening and closing operations:

$$I_{\text{opened}} = (I \ominus B) \oplus B \tag{9}$$

$$I_{\text{smooth}} = (I_{\text{opened}} \oplus B) \ominus B \tag{10}$$

where $B$ is a disk-shaped structuring element with radius $r = \max(3, \text{strength} \times 5)$, $\ominus$ denotes erosion, and $\oplus$ denotes dilation. Opening removes small isolated high-attention regions (noise), while closing connects nearby high-attention areas, effectively creating more coherent attention patterns.

*Adaptive Filtering:* Adaptive filtering combines Gaussian smoothing with local variance analysis to apply spatially-varying smoothing strength:

$$I_{\text{smooth}}(x, y) = w(x, y) \cdot I_{\text{gaussian}}(x, y) + (1 - w(x, y)) \cdot I(x, y) \tag{11}$$

where the adaptive weight $w(x, y)$ is computed based on local variance:

$$\text{Var}_{\text{local}}(x, y) = \frac{1}{|N|} \sum_{(i,j) \in N} (I(i, j) - \mu_N)^2 \tag{12}$$

$$w(x, y) = 1.0 - 0.7 \times \frac{\text{Var}_{\text{local}}(x, y) - \text{Var}_{\text{min}}}{\text{Var}_{\text{max}} - \text{Var}_{\text{min}}} \tag{13}$$

This method applies stronger smoothing in homogeneous regions (low variance) and preserves details in heterogeneous regions (high variance).

All methods incorporate a peak preservation mechanism that maintains the intensity of high-attention regions above the 90th percentile:

$$I_{\text{final}}(x, y) = \begin{cases} \alpha \cdot I(x, y) + (1 - \alpha) \cdot I_{\text{smooth}}(x, y) & \text{if } I(x, y) > P_{90} \\ I_{\text{smooth}}(x, y) & \text{otherwise} \end{cases} \tag{14}$$

where $\alpha = 0.7$ and $P_{90}$ is the 90th percentile threshold.

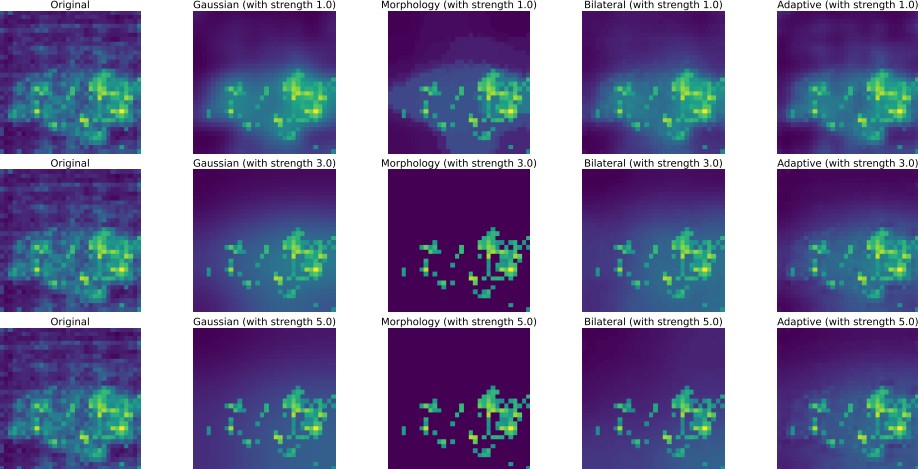

Figure 16: Comparison of filtering-based smoothing methods for local attention scores with varying smoothing strengths. Each row corresponds to different strength parameters (1.0, 3.0, 5.0). From left to right: original attention map, Gaussian filtering, morphological filtering, bilateral filtering, and adaptive filtering. Gaussian filtering provides uniform smoothing, morphological operations create connected regions through structural analysis, bilateral filtering preserves attention boundaries while smoothing homogeneous areas, and adaptive filtering intelligently varies smoothing strength based on local content variability. Higher strength values (bottom rows) produce increasingly smooth attention patterns, with adaptive filtering demonstrating superior performance in balancing detail preservation and spatial coherence.

**Interpolation Methods** *Radial Basis Function (RBF) Interpolation:* RBF interpolation constructs a smooth function $f(\mathbf{x})$ that passes through all given data points using a linear combination of radially symmetric basis functions:

$$f(\mathbf{x}) = \sum_{i=1}^{n} \lambda_i \phi(||\mathbf{x} - \mathbf{x}_i||) \tag{15}$$

where $\phi$ is the chosen kernel function (thin-plate spline in our implementation), $\mathbf{x}_i$ are the data points, and $\lambda_i$ are the interpolation weights. This method produces highly smooth results with natural spatial transitions, making it particularly effective for creating coherent attention regions.

*Cubic Interpolation:* Cubic interpolation uses piecewise cubic polynomials to create smooth transitions between data points. The method minimizes the total curvature while maintaining $C^2$ continuity, resulting in visually pleasing smooth surfaces. For 2D data, it employs bicubic interpolation:

$$f(x, y) = \sum_{i=0}^{3} \sum_{j=0}^{3} a_{ij} x^i y^j \tag{16}$$

This approach provides excellent balance between smoothness and computational efficiency.

*Linear Interpolation:* Linear interpolation creates piecewise linear surfaces between neighboring points using barycentric coordinates. While computationally efficient, it produces less smooth re-

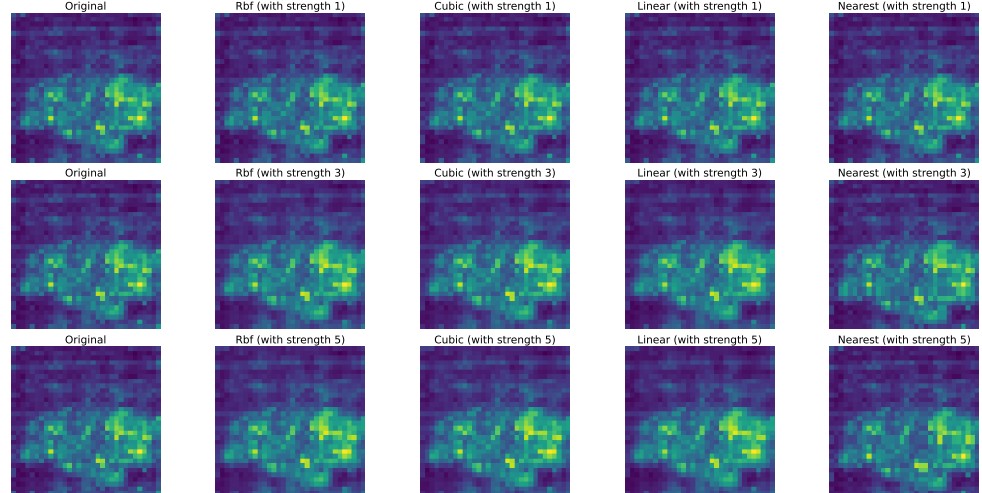

Figure 17: Comparison of interpolation-based smoothing methods for local attention scores. Each row shows results with different upsampling factors (1×, 3×, 5×). From left to right: original attention map, RBF interpolation, cubic interpolation, linear interpolation, and nearest neighbor interpolation. Higher upsampling factors (bottom rows) produce increasingly smooth and spatially coherent attention patterns, with RBF and cubic methods showing superior performance in connecting disjoint high-attention regions while preserving meaningful spatial structure.

sults compared to higher-order methods:

$$f(\mathbf{x}) = \sum_i w_i(\mathbf{x}) f_i \tag{17}$$

where $w_i(\mathbf{x})$ are the barycentric weights. This method preserves local features while providing moderate smoothing.

*Nearest Neighbor Interpolation:* The simplest interpolation method that assigns each interpolated point the value of its closest data point:

$$f(\mathbf{x}) = f_i \quad \text{where } i = \arg\min_j \|\mathbf{x} - \mathbf{x}_j\| \tag{18}$$

This method preserves sharp boundaries but provides minimal smoothing, serving as a baseline comparison.

All methods operate by first upsampling the $32 \times 32$ attention maps by a factor $k$ (where $k \in \{1, 3, 5\}$ in our experiments), applying the interpolation to create dense intermediate representations, then downsampling back to the original resolution. This process effectively fills gaps between high-attention regions and creates more spatially coherent attention patterns.

## C   EXPERIMENTS DETAILS

### C.1   TRAINING DETAILS

Throughout all training stages, we employ a resolution of $1024 \times 1024$ pixels, utilizing both publicly available datasets and our proprietary curated dataset. Training is conducted on $32 \times$ H100 GPUs. We adopt the AdamW optimizer Loshchilov & Hutter (2017) with hyperparameters $\beta_1 = 0.9$, $\beta_2 = 0.999$, weight decay of $1 \times 10^{-2}$, and $\epsilon = 1 \times 10^{-8}$. The learning rate is set to $1 \times 10^{-4}$ with gradient clipping at a maximum norm of 10. We use a batch size of 4 with gradient accumulation steps of 4, resulting in an effective batch size of 16.

For the first stage, we trained the model for 5,000 steps using 1M samples from JournerDB and PD-3M, which were re-annotated using InternVL-2.5-8B-MPO. Detailed information regarding the data can be found in Appendix C.2. In the second stage, we conducted full fine-tuning of the edit

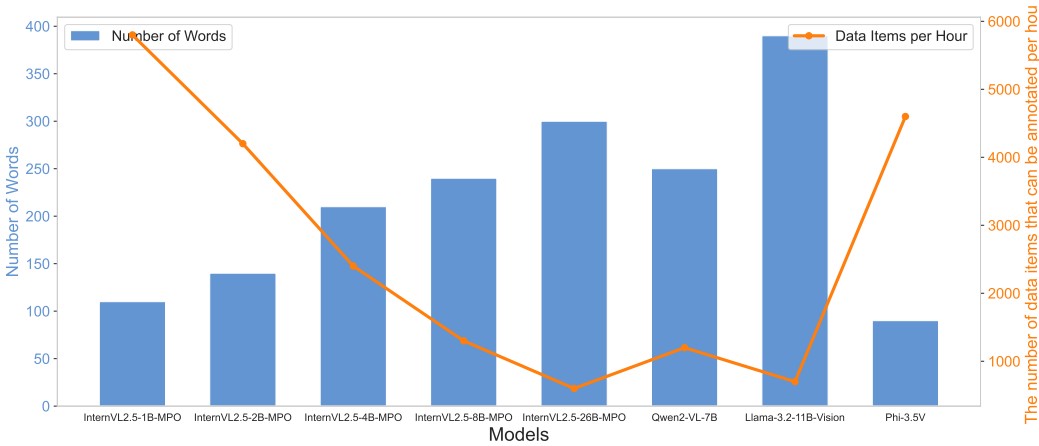

Figure 18: A comparison of captioning speed and caption length across common open-source VLMs. All models were evaluated using the same prompt and in $2\times$H100.

model on the complete 4M image editing dataset for 50,000 steps. The detailed data composition is provided in Appendix A.3. In the final stage, we performed additional training for 1,000 steps on approximately the top 12% of samples from the 4M dataset, selected based on their aesthetic quality scores (Schuhmann et al., 2022).

### C.2 RECAPTION

As mentioned in Section 3.3, we replace Meissonic's text encoder with Gemma2-2B-IT (Team et al., 2024a), necessitating the use of text-to-image datasets with extended captions to effectively leverage Gemma-2B's enhanced comprehension capabilities. To augment EditMGT's capacity for understanding and responding to complex linguistic instructions, we initially curate 1 million high-resolution samples with superior aesthetic scores from the JourneyDB (Pan et al., 2023) and PD-3M (Meyer et al., 2024) datasets.

Subsequently, we systematically re-caption our collected public dataset, which originally contained concise descriptions. The enhanced captions are deliberately crafted to provide comprehensive detail, with each image description spanning 180-320 characters. This refinement strategy aims to furnish richer contextual information and substantially improve the model's learning efficiency. Given the experimental validation, we utilize InternVL-2.5-8B-MPO for the annotation of our training data. To enhance data diversity, we generate three distinct captions per image and randomly select one during training, thereby augmenting the richness of our training corpus (Liu et al., 2024).

To evaluate the accuracy of the captions, we conducted experiments using CapsBench (Liu et al., 2024) and leveraged GPT-4o (Achiam et al., 2023) to assess the correctness of the captions generated by each model (Chow et al., 2025b). This rigorous evaluation process ensures that the selected model meets our high standards for both precision and reliability. The results of this evaluation can be found in Table 4. Additionally, we tested the captioning speed and caption length of several popular open-source VLMs, as illustrated in Figure 18.

We have also included some of the cases annotated in Table 4, as shown below.

**InternVL2.5-1B** (Chen et al., 2024c) **for Figure** 19. The image depicts a serene scene featuring an ancient stone bridge arching over a flowing river. The bridge, with its multiple arches, spans across the river, which is depicted in motion, creating a sense of tranquility. The riverbanks are lined with rocks and vegetation, adding to the natural beauty. In the background, a town with a prominent bell tower is visible, set against a backdrop of rolling hills and mountains under a dramatic sky. The sky is painted with warm hues of orange and yellow, suggesting either sunrise or sunset, enhancing the peaceful and picturesque atmosphere.

Table 4: The performance of some common VLMs on CapsBench (Liu et al., 2024). The indicators in the table are accuracy (%). The InternVL2.5 models are all MPO version.

| | InternVL2.5-1B | InternVL2.5-2B | InternVL2.5-4B | InternVL2.5-8B | InternVL2.5-26B | Qwen2-VL-7B | Llama-3.2-11B-Vision | Phi-3.5V |
|---|---|---|---|---|---|---|---|---|
| text | 64.39 | 56.82 | 61.36 | 59.09 | 61.36 | 60.61 | 38.64 | 27.27 |
| color | 55.67 | 58.42 | 60.14 | 58.08 | 62.54 | 58.42 | 60.48 | 37.11 |
| position | 40.25 | 41.49 | 41.08 | 43.15 | 46.89 | 45.23 | 43.57 | 48.96 |
| emotion | 62.79 | 61.63 | 59.30 | 55.81 | 62.79 | 56.98 | 58.14 | 60.47 |
| blur | 37.84 | 51.35 | 75.68 | 71.62 | 75.68 | 71.62 | 64.86 | 56.76 |
| artifacts | 14.29 | 11.43 | 17.14 | 20.00 | 20.00 | 8.57 | 2.86 | 57.14 |
| proper noun | 37.04 | 29.63 | 29.63 | 33.33 | 22.22 | 29.63 | 40.74 | 40.74 |
| entity shape | 46.46 | 40.40 | 41.41 | 40.40 | 38.38 | 39.39 | 41.41 | 52.53 |
| count | 67.88 | 72.26 | 73.72 | 71.53 | 73.72 | 67.15 | 67.88 | 31.39 |
| entity | 77.05 | 75.96 | 77.60 | 75.96 | 82.51 | 78.14 | 75.96 | 49.18 |
| relation | 50.34 | 51.02 | 57.82 | 51.70 | 62.59 | 55.10 | 52.38 | 41.50 |
| color palette | 75.65 | 78.26 | 80.87 | 85.22 | 80.87 | 80.00 | 64.35 | 67.83 |
| image type | 63.33 | 60.00 | 69.44 | 60.56 | 58.89 | 63.33 | 57.78 | 51.67 |
| color grading | 46.71 | 53.95 | 48.03 | 47.37 | 45.39 | 51.97 | 39.47 | 60.53 |
| relative position | 37.50 | 35.65 | 42.59 | 38.89 | 42.59 | 42.59 | 29.17 | 41.67 |
| general | 93.53 | 94.71 | 90.59 | 91.18 | 94.71 | 94.12 | 91.18 | 60.59 |
| entity size | 38.02 | 40.50 | 40.50 | 38.84 | 35.54 | 39.67 | 36.36 | 38.02 |

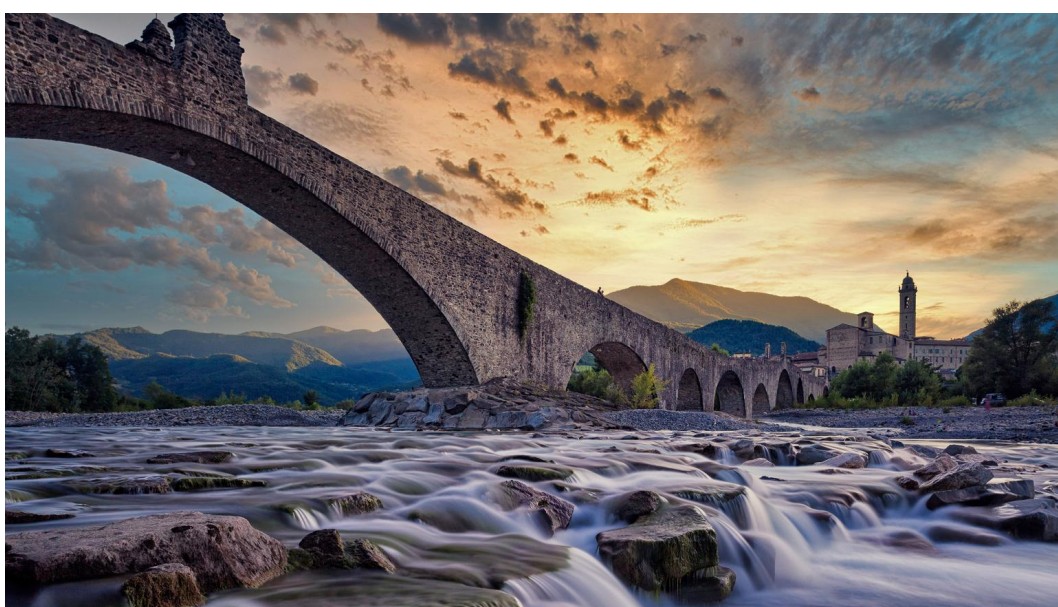

Figure 19: An example of VLMs recaptioning (Case i).

**InternVL2.5-2B** (Chen et al., 2024c) **for Figure** 19. A historic stone arch bridge spans a swiftly flowing river, its arches gracefully reflecting in the water below. The scene is captured during the golden hour, with the warm hues of the setting sun casting a soft glow over the landscape. In the background, a picturesque town with a prominent bell tower and rolling hills under a dramatic sky complete the picturesque view. The interplay of light and motion in the flowing water adds a dynamic element to the tranquil setting.

**InternVL2.5-4B** (Chen et al., 2024c) **for Figure** 19. The image captures a majestic stone bridge spanning a swiftly flowing river, with the water appearing as a dynamic blur due to a long exposure. The bridge, featuring multiple arches, stands as a testament to architectural heritage, its weathered stones contrasting with the vibrant natural surroundings. In the background, a picturesque town with a prominent bell tower is nestled among lush green hills, bathed in the warm, golden hues of a setting sun. The sky above is a dramatic canvas of orange and blue, with scattered clouds adding depth and texture. The lighting is soft yet dramatic, highlighting the textures of the bridge and the rippling water, while the depth of field ensures that the foreground, middle ground, and background are all sharply defined, creating a harmonious and captivating scene.

**InternVL2.5-8B** (Chen et al., 2024c) **for Figure** 19. The image captures a picturesque scene of an ancient stone bridge spanning a flowing river, set against a dramatic sunset sky. The artistic style is highly realistic, with a focus on the intricate textures of the bridge and the dynamic movement of the water. The lighting is a blend of natural and soft, with the warm hues of the sunset casting a golden glow across the sky and reflecting off the water, creating a serene and tranquil atmosphere. The

depth of field is shallow, with the foreground sharply focused on the flowing river and rocks, while the background features the bridge and distant mountains, slightly blurred to emphasize the depth. Notable color contrasts include the warm oranges and yellows of the sunset against the cooler blues and greens of the river and surrounding landscape. The texture of the stone bridge adds a tactile quality to the image, while the smooth, flowing water creates a sense of motion. The composition is balanced, with the bridge arching gracefully over the river, leading the eye towards the distant town and church tower. The reflections in the water enhance the overall atmosphere, adding a sense of calm and continuity. The weather conditions appear clear, with the vibrant colors of the sunset suggesting a peaceful evening. The image stands out with its harmonious blend of natural elements and human-made structures, capturing a moment of timeless beauty.

**InternVL2.5-26B** (Chen et al., 2024c) **for Figure** 19. The image captures a picturesque scene of an ancient stone bridge spanning a lively river, set against a backdrop of rolling mountains and a small town. The photograph is rendered in a highly realistic style, with meticulous attention to detail that brings out the textures of the stone bridge and the flowing water. The lighting is a blend of natural light, with the sun setting behind the mountains, casting a warm, golden glow across the sky and reflecting off the clouds. This creates a striking contrast with the cooler tones of the river and the shadows on the bridge, enhancing the depth and dimension of the scene. The depth of field is shallow, with the foreground – comprising the rushing river and large rocks – sharply in focus, while the background, including the bridge and the town, remains slightly blurred, drawing the viewer's eye through the image. The colors are rich and varied, with the warm hues of the sunset juxtaposed against the cool blues and greens of the landscape, creating a harmonious balance. The composition is masterfully arranged, with the bridge forming a strong diagonal line that leads the viewer's eye across the image. The river in the foreground adds a dynamic element, with its smooth, flowing water creating a sense of movement and energy. The background features a small town with a prominent bell tower, adding a touch of human presence to the otherwise natural scene. The sky, filled with scattered clouds, contributes to the dramatic effect, with the warm light breaking through and casting a serene, almost ethereal glow over the entire landscape. Overall, the image exudes a sense of tranquility and timelessness, capturing the beauty of nature and human craftsmanship in perfect harmony.

**Llama-3.2-11B-Vision** (Dubey et al., 2024) **for Figure** 19. In this captivating image, a majestic stone bridge spans a serene river, its arched structure and weathered appearance evoking a sense of history and timelessness. The bridge's intricate details, including the rough-hewn stones and the subtle wear of age, are beautifully rendered in the photograph. The river below flows gently, its crystal-clear waters reflecting the warm hues of the sky above. The water's surface is dotted with large rocks, adding texture and depth to the scene. In the background, a picturesque town unfolds, with buildings and trees nestled among rolling hills. The town's architecture is characterized by a mix of old and new, with some structures appearing centuries old and others more modern. The sky above is a kaleidoscope of colors, with shades of blue, orange, and yellow blending together in a breathtaking sunset. The clouds are scattered across the sky, adding a sense of movement and dynamism to the scene. The overall atmosphere of the image is one of tranquility and serenity, inviting the viewer to step into the peaceful world of the photograph. The image's style is realistic, with a focus on capturing the beauty of the natural world. The lighting is soft and warm, with the sun casting a gentle glow over the entire scene. The depth of field is shallow, with the bridge and surrounding landscape in sharp focus, while the background is softly blurred. The color palette is muted, with earthy tones dominating the scene. The texture of the stone bridge and the rough-hewn rocks adds a tactile quality to the image, while the flowing water creates a sense of movement and energy. The composition is well-balanced, with the bridge and town arranged in a harmonious relationship. The unique visual aspect of this image is the way the photographer has captured the play of light and shadow on the bridge's arches, creating a sense of depth and dimensionality.

**Qwen2-VL-7B** (Wang et al., 2024b) **for Figure** 19. This image captures a stunning sunset scene featuring a medieval stone bridge spanning a flowing river. The bridge, constructed with large, weathered stones, exhibits a classic Gothic architectural style, characterized by its arched design and robust construction. The lighting is dramatic, with the sun setting behind the mountains in the background, casting warm, golden hues across the sky and creating long shadows that add depth to the scene. The river below is turbulent, with rocks and boulders scattered along its banks, reflecting the vibrant colors of the sunset. The composition is balanced, with the bridge and river in the foreground, the mountains and sky in the middle ground, and the distant town with its church spire

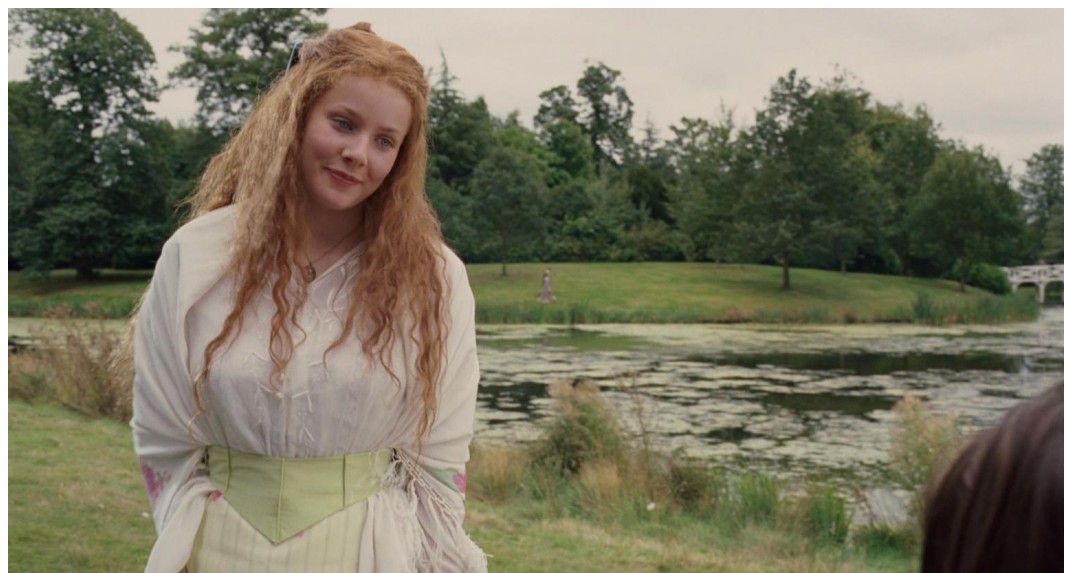

Figure 20: An example of VLMs recaptioning (Case ii).

in the background. The overall tone of the image is realistic, with a high level of detail in the textures of the bridge, rocks, and sky. The contrast between the dark, shadowed areas of the bridge and the bright, illuminated sky enhances the visual impact of the scene. The image is well-focused throughout, maintaining clarity in both the foreground and background elements.

**Phi-3.5V (AzureML, 2024) for Figure** 19. This image captures a stunning, historic stone bridge spanning a flowing river, set against a dramatic and picturesque landscape. The scene is bathed in the warm, golden hues of a setting sun, creating a serene and inviting atmosphere. The lighting is natural and soft, with the sun casting a gentle glow across the sky, highlighting the textures of the bridge and the river's surface.

**InternVL2.5-1B (Chen et al., 2024c) for Figure** 20. In this serene outdoor scene, a young woman with long, wavy red hair stands gracefully in a lush, green park. She is adorned in a delicate, off-white dress with a light green sash cinched at the waist, adding a touch of elegance to her appearance. Her gentle smile and poised demeanor suggest she is enjoying the peaceful surroundings. The background is a picturesque landscape featuring a tranquil pond dotted with lily pads, surrounded by a well-maintained lawn and a variety of trees that provide a lush canopy. A white bridge arches gracefully over the pond, adding a touch of elegance to the natural setting. The overcast sky casts a soft, diffused light over the scene, enhancing the calm and reflective atmosphere. In the foreground, the focus is on the woman, with the rest of the scene softly blurred, drawing attention to her and the tranquil environment around her. The composition is harmonious, with the foreground and background elements blending seamlessly, creating a sense of depth and tranquility. The overall style of the image is realistic, capturing the essence of a peaceful moment in a beautifully natural setting.

**InternVL2.5-2B (Chen et al., 2024c) for Figure** 20. A young woman with long, curly red hair stands gracefully in a serene, lush park. She wears a delicate, white, medieval-style dress with a light green corset, adding a touch of elegance to her appearance. The soft, natural light of the day illuminates her, casting gentle shadows and highlighting the intricate details of her attire. The background features a tranquil pond with lily pads, surrounded by verdant trees and a distant bridge, enhancing the peaceful ambiance. The scene is captured with a shallow depth of field, focusing on the woman while the background remains softly blurred, creating a harmonious and picturesque atmosphere.

**InternVL2.5-4B (Chen et al., 2024c) for Figure** 20. In this evocative scene, a young woman with flowing red hair stands gracefully in a lush, verdant landscape. She is dressed in a delicate, light-colored gown with a soft, pastel green sash cinched at her waist, adding a touch of elegance to her

serene presence. The background reveals a tranquil pond, its surface dotted with lily pads, and a distant bridge that adds a sense of depth and intrigue to the composition. The overcast sky casts a gentle, diffused light over the entire scene, enhancing the soft, dreamlike quality of the image. The woman's gentle smile and the natural beauty around her create a harmonious and peaceful atmosphere, inviting the viewer to lose themselves in this idyllic moment. The depth of field is shallow, with the woman in sharp focus against the softly blurred background, drawing attention to her as the central figure of this picturesque setting.

**InternVL2.5-8B** (Chen et al., 2024c) **for Figure** 20. The image is a realistic portrayal of a young woman with long, curly red hair, standing in a lush, green outdoor setting. The lighting is soft and natural, with an overcast sky diffusing the light evenly across the scene, creating a gentle and serene atmosphere. The depth of field is shallow, with the woman in sharp focus against a blurred background, emphasizing her presence. The color palette is dominated by cool tones, with the greenery providing a fresh contrast to her light-colored dress. The composition features the woman in the foreground, with a pond and trees in the middle ground, and a distant figure and bridge in the background, adding depth and context to the scene. The overall mood is calm and reflective, enhanced by the natural setting and the woman's gentle expression.

**InternVL2.5-26B** (Chen et al., 2024c) **for Figure** 20. The image portrays a serene, realistic outdoor scene with a young woman standing in a lush, green landscape. The photograph is lit with natural light, creating a soft and diffused effect, likely due to an overcast sky, which casts an even, cool-toned illumination across the scene. The woman, positioned in the foreground, is sharply focused, with her long, curly red hair and gentle smile adding warmth to the composition. She wears a light-colored, vintage-style dress with a green sash at the waist, contributing to the image's soft color palette. The background features a gently sloping grassy area leading to a pond covered with lily pads, enhancing the tranquil atmosphere. A bridge is visible in the distance, adding depth to the scene. The background is slightly blurred, creating a shallow depth of field that draws attention to the woman while maintaining a sense of space and setting. The natural textures of the grass, trees, and water contribute to the peaceful and idyllic mood of the image. The overall composition, with the woman in the foreground and the pond in the middle ground, leads the viewer's eye toward the softly focused background, creating a harmonious and inviting visual experience.

**Llama-3.2-11B-Vision** (Dubey et al., 2024) **for Figure** 20. In this image, a woman with long, wavy red hair stands in front of a serene pond, her gaze directed towards the camera. She wears a white blouse with a delicate floral pattern and a pale green corset, exuding a sense of elegance and refinement. Her hair is styled in a half-up, half-down manner, with a few loose strands framing her face. The woman's attire is complemented by a necklace featuring a small pendant, adding a touch of sophistication to her overall appearance. The background of the image features a picturesque pond, surrounded by lush green grass and trees, creating a tranquil atmosphere. A small bridge is visible on the right side of the image, adding depth and context to the scene.The lighting in the image is soft and natural, with the sun casting a warm glow over the entire scene. The color palette is muted, with earthy tones dominating the landscape. The overall mood of the image is one of serenity and peacefulness, inviting the viewer to step into the idyllic setting. The composition of the image is well-balanced, with the woman's figure positioned centrally, drawing the viewer's attention to her. The surrounding environment is carefully arranged to create a sense of harmony and balance, further enhancing the tranquil atmosphere of the image.

**Qwen2-VL-7B** (Wang et al., 2024b) **for Figure** 20. The image depicts a serene outdoor scene with a young woman standing in the foreground. She is dressed in a flowing, white, Victorian-style dress with intricate embroidery and a green corset belt. Her long, wavy red hair cascades down her shoulders, adding a touch of elegance to her appearance. The woman is looking slightly to her left, with a gentle, contemplative expression on her face. The background features a picturesque landscape with a pond covered in lily pads, surrounded by lush green grass and a variety of trees. The pond is situated in the middle ground, with a small bridge visible in the distance, adding to the idyllic setting. The lighting is soft and natural, with diffused sunlight filtering through the trees, creating a tranquil and peaceful atmosphere. The depth of field is shallow, with the woman in sharp focus while the background remains slightly blurred, drawing attention to her as the main subject. The overall style of the image is realistic, capturing the natural beauty of the scene with a high level of detail. The color palette is dominated by soft greens, blues, and whites, contributing to the calm and serene mood of the image. The composition is balanced, with the woman positioned centrally,

Table 5: Ablation Study GenEval benchmark (Ghosh et al., 2023) on $512 \times 512$. The pink highlighting indicates the final configuration adopted in our approach. "Attr." means Color Attribution.

| Model | Overall | Objects | | Counting | Colors | Position | Attr. |
|-------|---------|---------|-----|----------|--------|----------|-------|
| | | Single | Two | | | | |
| **Text Encoder** | | | | | | | |
| CLIP (Radford et al., 2021) | 0.44 | 0.95 | 0.63 | 0.11 | 0.78 | 0.08 | 0.10 |
| + Qwen2.5-0.5B (Team, 2024) | 0.45 | 0.97 | 0.65 | 0.08 | 0.84 | 0.08 | 0.09 |
| + T5-Large (Raffel et al., 2020) | 0.47 | 0.95 | 0.49 | 0.38 | 0.80 | 0.10 | 0.08 |
| + T5-XL (Raffel et al., 2020) | 0.48 | 0.97 | 0.66 | 0.14 | 0.83 | 0.09 | 0.18 |
| + T5-XXL (Raffel et al., 2020) | 0.49 | 0.97 | 0.76 | 0.19 | 0.78 | 0.13 | 0.14 |
| + Gemma1.1-2B (Team et al., 2024a) | 0.46 | 0.98 | 0.69 | 0.06 | 0.75 | 0.10 | 0.15 |
| + Gemma2-2B (Team et al., 2024b) | 0.42 | 0.91 | 0.56 | 0.08 | 0.80 | 0.13 | 0.08 |
| + Gemma2-2B-IT (Team et al., 2024b) | 0.50 | 0.99 | 0.78 | 0.16 | 0.82 | 0.10 | 0.11 |
| + Gemma3-1B (Gosthipaty et al., 2025) | 0.39 | 0.95 | 0.43 | 0.09 | 0.72 | 0.08 | 0.05 |
| + Gemma3-1B-IT (Gosthipaty et al., 2025) | 0.35 | 0.89 | 0.28 | 0.08 | 0.79 | 0.06 | 0.01 |
| + Llama3.2-1B (Meta AI, 2024) | 0.48 | 0.97 | 0.78 | 0.05 | 0.87 | 0.07 | 0.14 |
| + Wan Text (Wan et al., 2025) | 0.44 | 0.91 | 0.63 | 0.12 | 0.76 | 0.08 | 0.11 |
| Gemma-2B-IT (Team et al., 2024a) | 0.39 | 0.95 | 0.35 | 0.21 | 0.74 | 0.01 | 0.08 |
| **Connector Architecture** | | | | | | | |
| Linear | 0.47 | 0.95 | 0.49 | 0.38 | 0.80 | 0.10 | 0.08 |
| 2 layer MLP | 0.45 | 0.95 | 0.54 | 0.21 | 0.74 | 0.08 | 0.21 |
| 3 layer MLP | 0.30 | 0.91 | 0.09 | 0.08 | 0.74 | 0.00 | 0.00 |
| ELLA (Hu et al., 2024) | 0.38 | 0.95 | 0.34 | 0.09 | 0.70 | 0.09 | 0.06 |
| Qformer (Li et al., 2023) | 0.35 | 0.89 | 0.30 | 0.11 | 0.70 | 0.03 | 0.05 |

and the pond and trees framing the scene symmetrically. The image exudes a sense of nostalgia and tranquility, evoking a feeling of timelessness.

**Phi-3.5V** (AzureML, 2024) **for Figure** 20. The image captures a serene, pastoral scene with a young woman standing in a lush, green landscape. The style is realistic, with a soft, impressionistic artistic tone that enhances the tranquil atmosphere.

### C.3 LLM AS ENCODER

Our ablation studies were conducted on a $512 \times 512$ model, utilizing approximately 10% of the full training dataset. The experiments were uniformly performed using $8 \times$H100 GPUs, with training carried out for $10,000$ steps before evaluation. Mixed precision training was employed throughout the process. The training configuration included a batch size of 16, gradient accumulation steps of 8, and a learning rate of $1e - 4$.

For the **Text Encoder**, if we utilize a combination of features from two text encoders (such as Gemma2 and CLIP) to guide the process, we employ a single linear layer as the connector to ensure that the hidden size dimensions of both encoders remain consistent. Llama3.2-1B (Meta AI, 2024)'s pad token doesn't exist, so we use the EOS token to pad.

For the **Connector Architecture**, we employ CLIP+Gemma-2B as our text encoder. The input and output dimensions are 2304 and 768, respectively. A 2-layer MLP indicates that we utilize two linear layers with a GELU activation function in between. Similarly, a 3-layer MLP consists of three linear layers, each separated by a GELU activation function. Additionally, the dimensionality between the first and second linear layers is expanded to $2304 \times 4 = 9216$. For the Q-former, we follow the implementation of BLIP-2 (Li et al., 2023) and set the query emb length to be 6 and the layer number is 2. ELLA (Hu et al., 2024) introduces a time-step-aware Q-former (Li et al., 2023). Specifically, our configuration employs 3 layers, 8 heads, and a time-step controller with a dimensionality of 1024. Detailed experimental results are presented in Table 5.

### C.4 BASELINES DETAILS

We establish the models listed in Tables 2, 6, and 7 as our baseline methods. Our comparative evaluation encompasses four diffusion model-based approaches (InstructPix2Pix, UltraEdit, MagicBrush, and Null Text Inversion), one VAR-based method (VAREdit-8B), and two unified model approaches (OmniGen2 and GoT-6B). We strictly adhere to the default hyperparameters specified in the official

GitHub repositories or HuggingFace (Jain, 2022) implementations of these baseline models. The model architectures and key parameter configurations are detailed as follows:

- *InstructPix2Pix* (Brooks et al., 2023): This method leverages automatically generated instruction-based image editing datasets to fine-tune Stable Diffusion (Rombach et al., 2022b), thereby enabling instruction-conditioned image editing during inference without requiring any test-time optimization. In our experimental evaluation, we employ the following hyperparameters: `num_inference_steps=10` and `image_guidance_scale=1.0`.

- *UltraEdit* (Zhao et al., 2024): This model is trained on approximately 4 million instruction-based editing samples built upon the Stable Diffusion 3 () architecture. It supports both free-form and mask-based input modalities to enhance editing performance. For consistency across all experiments, we exclusively employ its free-form variant. We note that since UltraEdit is trained on the SD3 architecture, its performance metrics may not fully reflect the intrinsic improvements attributable to its specialized editing dataset. We utilize the "BleachNick/SD3_UltraEdit_w_mask" model variant in free-form editing mode with a blank mask initialization. The evaluation is conducted with hyperparameters `num_inference_steps=50`, `image_guidance_scale=1.5`, `guidance_scale=7.5`, and `negative_prompt=""` to maintain consistency with our experimental protocol. Inference is performed at $512 \times 512$ resolution, with estimated inference time of approximately 5 seconds at $1024 \times 1024$ resolution.

- *MagicBrush* (Kawar et al., 2023): MagicBrush presents a carefully curated editing dataset with comprehensive human annotations and fine-tunes its model on this dataset utilizing the InstructPix2Pix (Brooks et al., 2023) framework. During evaluation, we employ the following hyperparameters: `seed=42`, `guidance_scale=7`, `num_inference_steps=20`, and `image_guidance_scale=1.5`.

- *Null Text Inversion* (Mokady et al., 2023): This method performs inversion of the source image by leveraging the DDIM (Song et al., 2020a) sampling trajectory and executes semantic edits during the denoising process through the manipulation of cross-attention mechanisms between textual and visual representations. A critical constraint of Null Text Inversion is that attention replacement-based editing operations can only be applied to text prompts of identical token length. Consequently, when the source and target captions exhibit disparate lengths, we enforce length alignment by truncating the longer caption to match the shorter one. During evaluation, we configure the method with the following hyperparameters: `cross_replace_steps=0.8`, `self_replace_steps=0.5`, `blend_words=None`, and `equilizer_params=None`.

- *OmniGen2* (Wu et al., 2025b) is a unified multimodal generative model that demonstrates enhanced computational efficiency and modeling capacity. In contrast to its predecessor OmniGen v1, OmniGen2 employs a dual-pathway decoding architecture with modality-specific parameters for text and image generation, coupled with a decoupled image tokenization mechanism. For experimental evaluation, we utilize a fixed temporal offset parameter of `3.0`, set the text guidance scale to `5.0` and image guidance scale to `1.5`. The negative prompt is configured as `"(((deformed))), blurry, over saturation, bad anatomy, disfigured, poorly drawn face, mutation, mutated, (extra_limb), (ugly), (poorly drawn hands), fused fingers, messy drawing, broken legs censor, censored, censor_bar"`. All inference procedures employ the default 50-step sampling schedule.

- *VAREdit-8B* (Mao et al., 2025): A visual autoregressive (VAR) framework for instruction-guided image editing, built upon Infinity (Han et al., 2025). This approach reframes image editing as a next-scale prediction problem, achieving precise image modifications through the generation of multi-scale target features. We employ the following hyperparameters: classifier-free guidance scale `cfg=3.0`, temperature parameter `tau=0.1`, and random seed `seed=42`. We observe that VAREdit requires 16 seconds for the initial edit, with subsequent edits processed at 5 seconds per image.

- *GoT-6B* (Fang et al., 2025): GoT is a paradigm that enables visual generation and editing by transforming input prompts into explicit reasoning chains with spatial coordinates,

thereby facilitating vivid image generation and precise editing capabilities. We utilize the following parameter configuration: guidance scale `guidance_scale` $= 4.0$, image guidance scale `image_guidance_scale` $= 1.5$, and conditional image guidance scale `cond_image_guidance_scale` $= 3.0$.

## C.5 DETAILS ON BENCHMARKS

**Metrics and code**. For evaluation on the EMU Edit, MagicBrush, and AnyBench benchmarks, we adhere strictly to the MagicBrush evaluation protocol without modifications. Following established methodologies (Bai et al., 2024b; Zhang et al., 2024; Zhao et al., 2024), we utilize the L1 distance metric to quantify pixel-level discrepancies between generated outputs and ground truth images. Furthermore, we employ CLIP and DINO similarity scores to assess global semantic alignment with ground truth references, while CLIP-T evaluates text-image correspondence by computing alignment between local textual descriptions and CLIP embeddings of generated images. For evaluation on the GEdit-EN-full Benchmark, we just use the GPT.

**EMU-Edit-Test**. We observe that the original EMU-Edit (Sheynin et al., 2024) paper and dataset don't specify the versions of CLIP (Radford et al., 2021) and DINO (Zhang et al., 2022) used. To maintain consistency with other benchmarks, we follow the settings from the MagicBrush repository (Zhang et al., 2024), modifying only the evaluation dataset to EMU-Edit-Test.

**MagicBrush-Test**. MagicBrush is designed to evaluate both single-turn and multi-turn image editing capabilities of models. It provides annotator-defined instructions and editing masks, along with ground truth images generated by DALLE-2 (Ramesh et al., 2022), facilitating more effective metric-based assessment of model editing performance. However, the dataset exhibits inherent biases. During data collection, annotators are instructed to utilize the DALLE-2 image editing platform to generate edited images, rendering the benchmark biased toward images and editing instructions that the DALLE-2 editor can successfully execute. This bias may constrain the dataset's diversity and complexity. The baseline results presented in Table 1 of the main paper correspond to EMU-Edit (Sheynin et al., 2024). In our evaluation, we employ EditMGT's zero-shot masked editing capabilities.

**AnyBench**. To evaluate different tasks across various task categories, we conduct experiments on AnyBench, a carefully curated benchmark for unified and comprehensive assessment of instruction-based image editing capabilities, derived from the large-scale automatically constructed dataset AnyEdit. The benchmark encompasses 25 editing task categories. We exclude 8 vision-guided task categories and evaluate 14 task types across three major task categories: local, global, and implicit editing tasks.

**GEdit-EN-full Benchmark**. The benchmark comprises 610 instances, each consisting of a real image paired with an English editing instruction. Its primary objective is to evaluate the performance of existing editing algorithms in practical applications using authentic images and editing instructions. Model evaluation employs three metrics from VIEScore Ku et al. (2023): *Semantic Consistency (SQ)*: assesses the alignment between editing results and given editing instructions, with scores ranging from 0 to 10. *Perceptual Quality (PQ)*: evaluates image naturalness and the presence of artifacts, with scores ranging from 0 to 10. *Overall Score (O)*: computed based on the combined assessment of SQ and PQ metrics. Automatic evaluation is conducted using the GPT-4o model. The majority of data in Table 2 is sourced from GPT-Image-Edit Wang et al. (2025b), while OmniGen2 results are obtained from `https://github.com/VectorSpaceLab/OmniGen2/issues/45`.

## C.6 FIGURE DETAILS

**Comparison of open-sourced methods in Figure 1.** We conduct our experiments on a single H100 GPU with initially empty memory allocation, using a batch size of 1 throughout all evaluations. For inference time evaluation, we measure performance on $1024 \times 1024$ resolution images. The $512 \times 512$ results are extrapolated based on the computational scaling properties. The FLUX.1-Kontext-dev (Labs et al., 2025) contains 12B parameters and is evaluated using the default Hugging-Face configuration (28 inference steps, bfloat16 precision), achieving generation times of 26 seconds for $1024 \times 1024$ images and 8 seconds for $512 \times 512$ images. Bagel (Deng et al., 2025) employs the default configuration from its GitHub repository with bfloat16 precision, `num_timesteps=50`,

Table 6: **Comparison of Methods on AnyEdit-Test (Part 1)**. '-' indicates 'not applicable'.

| Method | Local | | | | | | | | |
|--------|-------|---------|-----|-------|------------|-----------------|--------|---------|----------|
| | remove | replace | add | color | appearance | material change | action | textual | counting |
| **InstructPix2Pix** (Brooks et al., 2023) | | | | | | | | | |
| CLIPim ↑ | 0.664 | 0.779 | 0.832 | 0.862 | 0.770 | 0.700 | 0.674 | 0.744 | 0.803 |
| CLIPout ↑ | 0.227 | 0.276 | 0.302 | 0.318 | 0.308 | - | 0.228 | 0.298 | - |
| L1 ↓ | 0.146 | 0.188 | 0.134 | 0.162 | 0.160 | 0.168 | 0.167 | 0.190 | 0.149 |
| DINO ↑ | 0.408 | 0.537 | 0.706 | 0.773 | 0.593 | 0.369 | 0.413 | 0.694 | 0.590 |
| **MagicBrush** (Zhang et al., 2024) | | | | | | | | | |
| CLIPim ↑ | 0.849 | 0.814 | 0.930 | 0.826 | 0.843 | 0.809 | 0.754 | 0.759 | 0.875 |
| CLIPout ↑ | 0.264 | 0.289 | 0.321 | 0.305 | 0.319 | - | 0.272 | 0.312 | - |
| L1 ↓ | 0.076 | 0.143 | 0.071 | 0.112 | 0.084 | 0.111 | 0.203 | 0.157 | 0.100 |
| DINO ↑ | 0.783 | 0.604 | 0.897 | 0.667 | 0.739 | 0.570 | 0.548 | 0.774 | 0.731 |
| **HIVE**$^w$ (Zhang et al., 2023b) | | | | | | | | | |
| CLIPim ↑ | 0.750 | 0.788 | 0.914 | 0.853 | 0.819 | 0.764 | 0.826 | 0.801 | 0.866 |
| CLIPout ↑ | 0.237 | 0.282 | 0.312 | 0.307 | 0.313 | - | 0.291 | 0.318 | - |
| L1 ↓ | 0.118 | 0.184 | 0.079 | 0.114 | 0.147 | 0.126 | 0.155 | 0.139 | 0.122 |
| DINO ↑ | 0.586 | 0.600 | 0.857 | 0.779 | 0.690 | 0.536 | 0.735 | 0.838 | 0.738 |
| **HIVE**$^c$ (Zhang et al., 2023b) | | | | | | | | | |
| CLIPim ↑ | 0.823 | 0.778 | 0.932 | 0.894 | 0.864 | 0.785 | 0.874 | 0.807 | **0.899** |
| CLIPout ↑ | 0.254 | 0.284 | 0.312 | 0.309 | 0.309 | - | 0.308 | 0.319 | - |
| L1 ↓ | 0.099 | 0.167 | 0.066 | 0.097 | 0.105 | 0.103 | 0.147 | 0.129 | 0.100 |
| DINO ↑ | 0.728 | 0.584 | 0.891 | 0.850 | 0.795 | 0.594 | **0.811** | 0.871 | 0.800 |
| **UltraEdit (SD3)** (Zhao et al., 2024) | | | | | | | | | |
| CLIPim ↑ | 0.806 | 0.805 | 0.925 | 0.851 | 0.817 | 0.764 | 0.827 | **0.854** | 0.880 |
| CLIPout ↑ | 0.262 | **0.295** | 0.323 | **0.320** | **0.320** | - | 0.292 | **0.344** | - |
| L1 ↓ | 0.087 | 0.151 | 0.072 | 0.091 | 0.100 | 0.108 | 0.158 | **0.127** | 0.089 |
| DINO ↑ | 0.709 | 0.615 | 0.867 | 0.791 | 0.729 | 0.522 | 0.724 | **0.890** | 0.764 |
| **Null-Text** (Mokady et al., 2023) | | | | | | | | | |
| CLIPim ↑ | 0.752 | 0.710 | - | 0.814 | 0.785 | - | 0.838 | 0.764 | - |
| CLIPout ↑ | 0.250 | 0.247 | - | 0.274 | 0.285 | - | 0.298 | 0.305 | - |
| L1 ↓ | 0.235 | 0.253 | - | 0.227 | 0.239 | - | 0.243 | 0.275 | - |
| DINO ↑ | 0.598 | 0.384 | - | 0.695 | 0.675 | - | 0.732 | 0.764 | - |
| **AnyEdit** (Yu et al., 2025) | | | | | | | | | |
| CLIPim ↑ | 0.851 | 0.853 | **0.946** | 0.896 | **0.877** | 0.811 | 0.873 | 0.763 | 0.898 |
| CLIPout ↑ | 0.265 | 0.292 | 0.322 | 0.313 | 0.309 | - | 0.306 | 0.303 | - |
| L1 ↓ | 0.103 | 0.123 | **0.052** | **0.061** | **0.051** | 0.084 | **0.145** | 0.136 | 0.088 |
| DINO ↑ | 0.785 | **0.688** | 0.921 | 0.855 | 0.840 | 0.602 | 0.782 | 0.800 | 0.819 |
| **EDITMGT (Ours)** | | | | | | | | | |
| CLIPim ↑ | **0.854** | **0.857** | 0.937 | **0.898** | 0.872 | **0.814** | **0.875** | 0.773 | **0.899** |
| CLIPout ↑ | **0.266** | 0.293 | **0.324** | 0.319 | 0.315 | - | **0.314** | 0.304 | - |
| L1 ↓ | **0.074** | **0.112** | 0.053 | 0.068 | 0.076 | **0.075** | 0.174 | 0.144 | **0.083** |
| DINO ↑ | **0.812** | 0.684 | **0.924** | **0.863** | **0.852** | **0.613** | 0.788 | 0.887 | **0.823** |

and `timestep_shift=3.0`. EditMGT utilizes a standard inference deployment configuration with 16 steps (EditMGT achieves optimal performance around 16 steps, with additional steps yielding no significant improvement). Under float32 precision, inference requires 4 seconds, while bfloat16 precision reduces this to 2 seconds with a total GPU memory consumption of 12.9 GB, where the model alone occupies 7.5GB of GPU cache. The hyperparameter configurations for OminiGen2 (Wu et al., 2025b), UltraEdit (Zhao et al., 2024), GoT-6B (Fang et al., 2025), and VAREdit-8B-1024 (Mao et al., 2025) during evaluation are detailed in Appendix C.4.

**Comparison of open-sourced datasets in Figure 1.** For the statistical analysis of data types, categories exceeding 8 types are uniformly plotted within the $8 - 9$ range on the visualization, where the vertical position of data points' centroids still preserves the relative ordering of category counts. For resolution analysis, we employ a coarse subsampling approach to compute the mean resolution of the edge, which serves as the x-axis values in our plots.

**Details for Figure 4.** We randomly sampled 50 data points from the Gedit Bench En part. The semantic score reported in the figure corresponds to the overall score. For the L1 score calculation,

Table 7: **Comparison of Methods on AnyEdit-Test (Part 2)**. '-' indicates 'not applicable'.

| | global | | | implicit | |
| --- | --- | --- | --- | --- | --- |
| | background | tone transfer | style change | implicit | relation |
| **InstructPix2Pix** (Brooks et al., 2023) | | | | | |
| CLIPim ↑ | 0.680 | **0.860** | 0.702 | 0.762 | 0.826 |
| CLIPout ↑ | 0.259 | 0.304 | - | - | 0.288 |
| L1 ↓ | 0.221 | **0.098** | 0.221 | 0.212 | 0.167 |
| DINO ↑ | 0.411 | 0.804 | 0.354 | 0.538 | 0.577 |
| **MagicBrush** (Zhang et al., 2024) | | | | | |
| CLIPim ↑ | 0.739 | 0.789 | 0.664 | 0.819 | 0.910 |
| CLIPout ↑ | 0.268 | 0.287 | - | - | 0.280 |
| L1 ↓ | 0.233 | 0.213 | 0.252 | 0.189 | 0.109 |
| DINO ↑ | 0.529 | 0.657 | 0.292 | 0.622 | 0.800 |
| **HIVE$^w$** (Zhang et al., 2023b) | | | | | |
| CLIPim ↑ | 0.764 | 0.816 | 0.706 | 0.784 | 0.858 |
| CLIPout ↑ | 0.280 | 0.293 | - | - | 0.284 |
| L1 ↓ | 0.202 | 0.175 | 0.212 | 0.202 | 0.119 |
| DINO ↑ | 0.635 | 0.719 | 0.383 | 0.572 | 0.697 |
| **HIVE$^c$** (Zhang et al., 2023b) | | | | | |
| CLIPim ↑ | **0.822** | 0.833 | 0.705 | 0.809 | **0.914** |
| CLIPout ↑ | 0.294 | 0.293 | - | - | 0.284 |
| L1 ↓ | 0.177 | 0.182 | 0.401 | 0.180 | 0.093 |
| DINO ↑ | **0.777** | 0.748 | 0.202 | 0.627 | **0.829** |
| **UltraEdit (SD3)** (Zhao et al., 2024) | | | | | |
| CLIPim ↑ | 0.790 | 0.795 | 0.730 | 0.825 | 0.887 |
| CLIPout ↑ | 0.293 | 0.301 | - | - | 0.281 |
| L1 ↓ | 0.181 | 0.184 | 0.208 | 0.176 | 0.093 |
| DINO ↑ | 0.701 | 0.709 | 0.448 | 0.642 | 0.764 |
| **Null-Text** (Mokady et al., 2023) | | | | | |
| CLIPim ↑ | 0.755 | 0.750 | - | - | - |
| CLIPout ↑ | 0.285 | 0.269 | - | - | - |
| L1 ↓ | 0.251 | 0.289 | - | - | - |
| DINO ↑ | 0.617 | 0.608 | - | - | - |
| **AnyEdit** (Yu et al., 2025) | | | | | |
| CLIPim ↑ | 0.819 | 0.836 | 0.710 | 0.825 | 0.908 |
| CLIPout ↑ | **0.300** | 0.302 | - | - | **0.289** |
| L1 ↓ | **0.169** | 0.115 | **0.192** | 0.169 | **0.091** |
| DINO ↑ | 0.744 | **0.811** | 0.385 | 0.643 | 0.822 |
| **EDITMGT (Ours)** | | | | | |
| CLIPim ↑ | 0.815 | 0.837 | **0.746** | **0.831** | 0.904 |
| CLIPout ↑ | 0.297 | **0.305** | - | - | **0.289** |
| L1 ↓ | 0.178 | 0.130 | 0.258 | **0.162** | 0.094 |
| DINO ↑ | 0.753 | 0.809 | **0.464** | **0.654** | 0.827 |

since images processed through VQ-VAE (Crowson et al., 2022) exhibit inherent L1 reconstruction error (approximately 0.05 as measured in our experiments), we treat the image with $\lambda = 1$ as the reference baseline for computing L1 scores.

Table 8: Specific design choices employed by masked generative Transformers (MGTs) are presented in this overview. We adopt a definitional form of sampling that is consistent with DMs, akin to EDM (Karras et al., 2022). Let $N$ denote the number of sampling steps, and the sequence of time steps is $\{t_0, \cdots, t_N\}$, where $\sigma_{t_N} = 0$.

| | | DM (Song et al., 2020b) | ARM (Sun et al., 2024) | MGT (Bai et al., 2024b) |
|---|---|---|---|---|
| **Definition** | | | | |
| TimeStep | $t_{0 \le i \le N}$ | $t = 1 + \frac{i}{N}(\epsilon - 1)$ (VP-SDE & flow matching) $i/N$ (EDM) | N/A (next-token prediction) | $i/N$ (non-ar token prediction) |
| Noise Schedule | $\sigma_t$ | $\sqrt{e^{a^2 t + bt} - 1}$ (VP-SDE (Song et al., 2020b)) $t$ (flow matching (Liu et al., 2022)) $(\sigma_{max}^{\frac{1}{\rho}} + t(\sigma_{min}^{\frac{1}{\rho}} - \sigma_{max}^{\frac{1}{\rho}}))^\rho$ (EDM (Karras et al., 2022)) | N/A, and predicts one token per iteration | $\cos\left(\frac{\pi t}{2}\right)$ |
| Network Architecture | $f_\theta$ | U-Net or Transformer (encoder only) | Transformer (decoder only) | Transformer (encoder only) |
| Coding Form | $Q(ez \mid ex)$ | VAE (Kingma, 2013) (continuous) | VQ-VAE (Van Den Oord et al., 2017) (discrete) | VQ-VAE (Van Den Oord et al., 2017) (discrete) |
| **Inference** | | | | |
| Sampling Paradigm $p(ez_i \mid \prod_{j<i} ez_j)$ | | DDPM (Ho et al., 2020), Euler (Song et al., 2020b), Classifier-free Guidance (Ho & Salimans, 2022), Z-Sampling (Bai et al., 2024c), et al. | Autoregressive ($ez_i$ denotes a token) | MaskGIT's Sampling (Chang et al., 2022) ($ez_i$ denotes all masked tokens) |
| Improved Probability Distribution | | N/A | $\arg\max_i \frac{\log(\epsilon)}{ep}$, where $ep$ is the logit and $\epsilon \sim \mathcal{U}[e0, e1]$ | $\arg\max_i \frac{\log(\epsilon)}{ep}$, where $ep$ is the logit and $\epsilon \sim \mathcal{U}[e0, e1]$ |
| **Editing** | | | | |
| Method | | Additional Channels Additional Adapter Hidden States Addition Denoising Inversion | Token Arrangement In-context | EDITMGT (Ours) |

# D MORE RELATED WORK

Existing image editing models are primarily adapted from text-to-image generative models, leveraging their robust textual comprehension capabilities and image generation capacities (Huang et al., 2025b; Chow et al., 2025b; Chen et al., 2025). Based on the underlying generative framework, these models can be classified into three primary categories: Diffusion Models (DM) (Podell et al., 2023; Esser et al., 2024; Song et al., 2020a), Autoregressive Models (ARM) (Sun et al., 2024; Li et al., 2024; Pan et al., 2024; Deng et al., 2025), and Masked Generative Transformers (MGT) (Bai et al., 2024b; Chang et al., 2023; 2022). Based on recent literature (Shao et al., 2024), we provide a comprehensive summary of the definitions, inference methods, and associated editing techniques for DM, ARM, and MGT, as outlined in Table 8.

**DM-based Editing** . The diffusion models (DMs) has emerged as the predominant framework for both text-to-image generation and image editing tasks in contemporary research (Yan et al., 2025; Liu et al., 2025; Huang et al., 2025a; Yang et al., 2025b; Shi et al., 2024; Peebles & Xie, 2023; Cai et al., 2025b;b; Wang et al., 2025a; Jiang et al., 2025; Wang et al., 2025b). Prompt-to-Prompt (Hertz et al., 2022) is an early image editing approach that operates by injecting the attention maps of the input caption into those of the target caption. Null-Text Inversion (Mokady et al., 2023) inverts the source image to the null-text embedding for editing, eliminating the need for original captions. GLIDE (Nichol et al., 2021) and Imagen Editor (Wang et al., 2023) fine-tuning the model to take channel-wise concatenation of the input image and mask. Blended Diffusion (Avrahami et al., 2022; 2023) blends the input image in the unmasked regions in the diffusion step. Meanwhile, instruction-based image editing has been introduced as a user-friendly method for image editing. InstructPix2Pix (Brooks et al., 2023) extends the original text-to-image generation model to an image editing model by incorporating an additional channel in a U-Net architecture (Ronneberger et al., 2015) to introduce the original pre-edit image. MGIE (Fu et al., 2023) jointly trains a DM and a MLLM Liu et al. (2023a;b) to enhance the editing model's capability in comprehending textual instructions. Subsequent approaches have primarily followed the same line of thought, which can be broadly categorized into four main groups: additional channels (Brooks et al., 2023; Kawar et al., 2023; Zhao et al., 2024; Yu et al., 2025; Zhou et al., 2025; Han et al., 2024; Hu et al., 2025; Li et al., 2025), additional adapter(Ye et al., 2023; Mou et al., 2023; Feng et al., 2024; Ye et al., 2023; Mou et al., 2024; He & Yao, 2025), hidden states addition (Zhao et al., 2023; Zhang et al., 2023a; Labs, 2024; Zhang et al., 2023b) and denoising inversion (Mokady et al., 2023; Tang et al., 2024; Avrahami et al., 2022; Rout et al., 2024; Xu et al., 2024; Wang et al., 2024a; Sheynin et al., 2024; Kulikov et al., 2024; Zhu et al., 2025).

**ARM-based Editing.** Make-a-scene (Gafni et al., 2022) handles text tokens, scene tokens and image tokens with a autoregressive transformer. VQGAN-CLIP (Crowson et al., 2022) introduces a method for text-conditioned image generation and editing (Xu et al., 2025b). The editing mechanism stems from the fusion of VQGAN's image synthesis capabilities with CLIP's ability to steer image transformations through textual guidance. This framework permits users to modify existing images or synthesize novel ones by altering stylistic attributes, introducing new elements, or transforming specific regions while maintaining visual consistency. In comparison, Make-A-Scene (Gafni et al., 2022) advances this paradigm by integrating scene layouts (in the form of segmentation maps) alongside textual conditioning. This extension enables finer-grained control over both structural composition and content generation, particularly facilitating localized editing operations. Whereas Make-A-Scene provides dual control over semantic content and spatial configuration, VQGAN-CLIP primarily facilitates open-ended, text-guided creative manipulation. EditAR (Mu et al., 2025) represents the first model to leverage an ARM architecture for image editing by encoding the original image as an in-context input for an autoregressive model and subsequently predicting the edited output. Uniworld (Lin et al., 2025) is a unified generative model that leverages high-resolution semantic encoders to achieve state-of-the-art performance in image understanding, generation, and manipulation tasks with remarkable data efficiency. Bagel (Deng et al., 2025), as a state-of-the-art unified model for multimodal understanding and generation, can naturally leverage contextual images to generate edited images combined with textual language. However, due to its autoregressive generation approach, the model lacks explicit spatial alignment, resulting in imperfect pixel-level consistency between the generated images and the original ones. NEP (Wu et al., 2025c) employs autoregressive image generation to selectively regenerate only the regions requiring modification, thereby preventing unintended alterations to non-edited areas while enhancing both computational efficiency and editing fidelity. Qwen-Image (Wu et al., 2025a) is currently the strongest AR-based editing model which combines Qwen2.5-VL semantic features with VAE reconstructive latents in an MMDiT backbone and is trained with curriculum and multi-task objectives to deliver consistent edits.

**MGT-based Editing** Current research leveraging the MGT (Masked Generative Transformer) architecture for text-to-image editing remains relatively limited, with applications primarily confined to image inpainting (Ko & Kim, 2023; Kim et al., 2023) and interpolation (Ma et al., 2024). To the best of our knowledge, EditMGT is the first MGT-based framework designed for general image editing. By exploiting the inherent semantic information encoded in its attention mechanisms and the token-flipping nature of the generation process, we introduce multi-layer attention consolidation along with a region-hold sampling technique to explicitly mitigate the issue of editing leakage.

## E    BROADER IMPACT

### E.1    IMPACT

The broader impact of EditMGT carries both potential benefits and risks upon deployment and release. Some considerations are unique due to the multimodal nature of edit model while others reflect challenges common to image creation environments. Below, we outline risks and mitigation strategies for its release.

**Hallucination.** Similar to other editing models (Yu et al., 2025; Brooks et al., 2023), our approach extends and fine-tunes text-to-image generation models to obtain editing capabilities, which introduces potential hallucination issues (Ji et al., 2023). Analogous to existing methods, models trained on EditMGT may produce outputs that deviate from user intentions or specified input conditions. This phenomenon raises significant concerns, particularly in commercial image applications where purchasing decisions rely on accurate visual representations, given that user requirements and expression modalities exhibit inherent variability.

**Biases.** Training data biases may propagate through EditMGT implementations, manifesting in both visual feature extraction and linguistic interpretation components. This propagation can yield biased retrieval results and inequitable representations across diverse cultural contexts. Multilingual processing introduces additional bias vectors through language alignment mechanisms, as demonstrated by (Gallegos et al., 2024).

**Ethical Considerations.** This work presents no significant ethical concerns. Our open-source data and model releases adhere to established corporate policies and industry standards governing intellectual property rights and data distribution practices (Coburn & Turner, 2012).

**Expected Societal Implications.** The compact editing model with 960MB parameters can provide significant benefits to the image creation community, particularly in resource-constrained scenarios. However, challenges remain in ensuring fairness across linguistic and cultural boundaries. Strong ethical standards and ongoing evaluation are essential for maximizing positive impact. These issues are not unique to our method but are prevalent across different techniques for image editing. Despite these challenges, we believe the benefits significantly outweigh the potential limitations, enabling continued investigation and improvement of image editing models while engaging the community in developing superior approaches. Moreover, the release of MODELNAME can foster novel applications and research directions, contributing to the advancement and responsible deployment of image editing technologies in resource-limited environments.

## E.2 LIMITATIONS

**Limited Training Scale:** Due to computational constraints, our model was trained on a dataset containing only 5M samples. This limited scale may adversely impact the generalization capabilities compared to models trained on larger-scale datasets, potentially restricting the model's performance across diverse scenarios.

**Inherited Model Deficiencies:** The underlying text-to-image generation models exhibit inherent limitations, occasionally producing images with cartoon-like stylistic artifacts or other visual distortions in the generated outputs. These limitations are not attributable to our proposed methodology, but rather stem from the constraints of existing state-of-the-art masked generative transformer (MGT) architectures. Future research directions could address these issues through the development of more robust foundational text-to-image

## E.3 REPRODUCIBILITY STATEMENT

We adhere to standard baseline configurations from established evaluation benchmarks or the original testing protocols of the respective models. All implementation details of our approach are provided in Appendix C. In compliance with the ICLR Reproducibility Requirements, we will release our data and code under an open-access license, along with comprehensive documentation to facilitate the exact replication of the key experimental results reported in this work.

## E.4 DECLARATION

This work is conducted exclusively for academic research purposes and contains no commercial elements. Our dataset is derived from publicly available sources, and the annotation models utilized are based on open-source frameworks. We are committed to upholding intellectual property rights and copyright protections. Should any visual content presented in this paper raise copyright concerns, we will promptly address such issues by removing the relevant materials. We plan to open-source our editing dataset and model weights under the **CC BY-NC 4.0** (Creative Commons Attribution-NonCommercial 4.0) license to facilitate future research endeavors.

# F THE USE OF LARGE LANGUAGE MODELS (LLMS)

In this work, large language models (LLMs) are employed in three limited capacities: (i) to refine the writing and enhance the linguistic clarity of the manuscript; (ii) to generate instructions during the construction process of CrispEdit-2M, where LLMs serve both as generative agents and supervisory filters for quality control.

