# OpenReview forum: "EditMGT: Unleashing Potentials of Masked Generative Transformers in Image Editing"
_ICLR.cc/2026/Conference — ICLR 2026 Conference Desk Rejected Submission_

### Official Review · Reviewer_oz2U · 2025-10-17

**Soundness:** 3
**Presentation:** 3
**Contribution:** 3
**Rating:** 6
**Confidence:** 4

**Summary:**

This paper introduces EditMGT, an image editing framework based on Masked Generative Transformers (MGTs) instead of diffusion models. The method uses attention injection to transfer structure from the source image, multi-layer attention consolidation to localize edits, and region-hold sampling to preserve unedited areas. It also introduces CrispEdit-2M, a large high-resolution dataset for editing tasks. Experiments on multiple benchmarks show that EditMGT achieves strong image quality and instruction-following performance while running much faster and with fewer parameters than diffusion-based approaches.

**Strengths:**

- The paper presents an innovative use of Masked Generative Transformers for image editing, offering a fresh alternative to diffusion models.
- The attention injection mechanism enables efficient conditioning without additional parameters, making the method lightweight and practical.
- The combination of attention consolidation and region-hold sampling provides precise edit localization and strong preservation of unedited areas.
- Experiments across several benchmarks show that EditMGT delivers competitive or superior results with significantly faster inference speed.
- The introduction of CrispEdit-2M contributes a valuable high-resolution dataset that can benefit future research on transformer-based editing.

**Weaknesses:**

- The paper lacks detailed ablation studies that isolate the effect of each component on the final performance.
- The robustness of the approach is not deeply analyzed, especially for challenging cases such as small or overlapping objects, fine-grained texture edits (e.g., 'add a small cat logo on the cup'), or ambiguous text prompts (e.g., 'make it blue' -- in a scene we have multiple (two or three) objects) where localization may fail.
- Some quantitative metrics, such as L1, show inconsistent improvements, suggesting room for better fidelity evaluation.
- The dataset description could be more thorough in addressing licensing terms, data provenance, and potential biases.

**Questions:**

1. Could the authors include ablation results showing the contribution of attention injection, consolidation, and region-hold sampling individually?
2. How are the λ and γ parameters selected, and how sensitive is model performance to their values?
3. What are the main failure modes when edits are small, ambiguous, or involve overlapping objects?
4. Have the authors considered using region-aware metrics to better assess edit preservation quality?
5. Could the authors provide more detail on the licensing and bias mitigation procedures used for CrispEdit-2M?


Typo:
- L403: 960MB or 960M

---

### Official Review · Reviewer_1D5h · 2025-10-24

**Soundness:** 4
**Presentation:** 3
**Contribution:** 3
**Rating:** 8
**Confidence:** 4

**Summary:**

This paper proposes a novel image editing framework based on masked generative transformers. In most image editing tasks, there exist image regions that remain unchanged before and after editing, where masked generative transformers can handle such situations well. The authors make good use of the properties of MGTs and design novel inference-time methods to enhance preservation of unedited regions. In this paper, a large-scale, high-quality dataset for image editing is constructed. Experimental results demonstrate the efficiency and effectiveness of the proposed framework. With less than 1B parameters, the proposed EditMGT achieves comparable performance against models with 6B or 8B parameters.

**Strengths:**

This paper employs masked generative transformers for image editing. The nature of MGTs, which process local patches for image generation, allows it to better preserve unedited regions in image editing tasks. The idea is logical, and the framework is pioneering. Experimental results also demonstrate the potential of this method: it achieves state-of-the-art performance comparable to large models while requiring much fewer parameters.

**Weaknesses:**

- The absence of an ablation study makes it unclear how much the proposed inference-time techniques, i.e., attention consolidation and region-hold sampling, contribute to efficiency and final metrics. I am concerned that these methods may introduce additional computational overhead.

- There is an issue of fairness in the comparative experiments. The paper mentions the use of CrispEdit-2M, a specially collected high-quality and high-resolution dataset, which may have significantly contributed to the performance gains.

**Questions:**

- Regarding region-hold sampling: What would be the impact if it were removed? Given that the model is trained on numerous source/target image pairs, preserving unedited regions should inherently be learned during training. Why is this extra inference-time mechanism necessary, especially since it may introduce additional computational overhead?

- Concerning cross-attention scores: The paper states that cross-attention maps contain rich semantic information. Could this property be explicitly enhanced during training? For instance, by incorporating masks of edited regions to supervise the distribution of attention scores? Would such a strategy improve the accuracy of region-hold sampling?

- On dataset contribution: The method uses the specially collected high-resolution CrispEdit-2M dataset. If such additional high-quality data were not used, would the proposed EditMGT framework still demonstrate clear advantages?

---
Typos:
* There are repeated citations in the references. Line 864-870.
* Line 57: "in this domain is (DMs)", missing words.
* Line 2224 "Stable Diffusion 3 () architecture", missing words.

---

### Official Review · Reviewer_Ty38 · 2025-10-24

**Soundness:** 3
**Presentation:** 3
**Contribution:** 2
**Rating:** 6
**Confidence:** 4

**Summary:**

This work introduces a novel method for instruction based image editing. The main claim of the paper is to be the first image editing method based on masked generative transformers while all competitors are diffusion based. The use of a masked generative transformer in turns means that the proposed model can have some nice properties like a more explicit localization of the edits and strong performance even with a modest parameter count. The authors also introduce a new dataset for training instruction based image editing models. Combining the new datasets with the proposed architecture they are able to train a small model (<1B parameters) that is competitive with SOTA for a fraction of the compute budget.

**Strengths:**

+ The proposal takes a text-to-image masked generative transformer and repurpose it for the task of instruction based image editing. The guidelines and best practices identified in the paper are likelly reusable when/if new and better masked generative transformers become available

+ The result of this work is an image editing model that is significantly cheaper to run and requires a fraction of the memory compared to most competitors. It achieves so without sacrificing too much the quality of edits.

+ The authors also contribute a large scale image editing training set obtained by running 2 heavy models (Flux.1 Kontext and Step1X-Edit v1.2) on ~2M images and applying several data cleaning heuristics. This in itself is a sizable contribution as it can enrich the community with more high quality training data.

**Weaknesses:**

## Major

a. **Entanglement of contribution/missing ablations**: the paper has 2 main contributions: a novel dataset for instruction based image editing and a method to turn masked generative transformers into image editing models. When the two are combined the paper shows that they are able to generate a model which is small but competitive at the task, however it is not clear what are the relative contributions of data vs architecture wrt this achievement. In particular the paper would have benefitted from either having an existing  diffusion based image editing architecture trained on the mixture of datasets that this work uses (including the new one) or training the new MGT based architecture on one of the existing datasets to compare it to a DM based solution. As it stands as a reader I cannot safely say whether using MGT is as good or better than a comparable DM since architectures are trained on very different mixtures. This dilutes the value of the contribution of a new architecture as it is unclear whether the good performance are merits of the design choices in model development or in data creation (or both).

b. **Need to identify “objects” in the edit instruction to achieve localization**: according to my understanding of lines 292-294 the method needs to identify “objects” in the edit instruction to extract the corresponding cross-attention masks and enforce edits to be local. This is either a requirement for some additional input compared to competitors or it is implemented using heuristics that might be brittle and fail in corner cases. This part would need to be clarified in the paper.

## Minor

C. **Architectural contributions and novelty claims**: The paper claims to be the first method to propose a MGT based method for image editing. I would suggest the authors rephrase this claim since even the original Muse paper was already showing examples of image editing although it was not the main point of the work (for examples see fig. 2 in [1]). From the point of view of architectural contributions most changes are 1 to 1 port of ideas used in the DM community adapted for MGT models: concatenating a “conditioning image” tokens in the model is somewhat equivalent to concatenating images channel wise as done in DM models like IP2P [2], relying on cross attention to achieve localized edits is also widely exploited in the DM community on top of already trained models for both generation [3] and edits [4], Eq. 2 with \gamma controlling the intensity of the source image conditioning sounds to me like the MGT equivalent of classifier free guidance (btw can this model use classifier free style inference to further improve quality?). So the core ideas are not novel, but what is novel is the way these techniques have been rethought in a MGT based model. I would have appreciated the work to be more transparent on this aspect.

D. **Several hyperparameter to tune**: The method has several degrees of freedom at test time and the paper is in my opinion not extremely clear on whether they have been kept fixed for all the experiments or they require tuning per dataset. Even crucial information like the value of \gamma in Eq. 2 and the value of \delta in the “region hold sampling” are not specified and it is unclear whether they are fixed or not at inference.  Same for the selection of layers for attention aggregation: the only motivation the authors provided for picking these layers is represented by Fig. 9, this is somewhat subjective, I would have appreciated a more experimentally driven approach.

E. **Visual presentation**: I appreciate the authors trying to have more visually pleasing representation for their results, but in some cases this goes a bit too much into the pretty graphic territory removing the scientific relevance. For example:  Fig. 4 left should report the value used for \lambda_1, \lambda_2 etc, Fig. 5 (a) and (b) has no scale across any of the axes. Fig. 2 is not using any of the terminology introduced in the method section which makes the mapping hard.


## Reference

1. [Chang, Huiwen, et al. "Muse: Text-to-image generation via masked generative transformers." arXiv preprint arXiv:2301.00704 (2023).](https://arxiv.org/pdf/2301.00704)

2. [Brooks, Tim, Aleksander Holynski, and Alexei A. Efros. "Instructpix2pix: Learning to follow image editing instructions." Proceedings of the IEEE/CVF conference on computer vision and pattern recognition. 2023.](https://arxiv.org/pdf/2211.09800)

3. [Hertz, Amir, et al. "Prompt-to-prompt image editing with cross attention control." arXiv preprint arXiv:2208.01626 (2022).](https://arxiv.org/pdf/2208.01626)

4. [Simsar, Enis, et al. "Lime: localized image editing via attention regularization in diffusion models." 2025 IEEE/CVF Winter Conference on Applications of Computer Vision (WACV). IEEE, 2025.](https://arxiv.org/pdf/2312.09256)

5. [Ho, Jonathan, and Tim Salimans. "Classifier-free diffusion guidance." arXiv preprint arXiv:2207.12598 (2022).](https://arxiv.org/pdf/2312.09256)

**Questions:**

1. How do you set \gamma in Eq.2 during training? If it’s sampled, do you still train a pure text-to-image branch?

2. Is the Flux.1 Kontext used to generate the CrispEdit 2M dataset the same to which the author compares to in Tab. 2? Also why neither Flux.1 Kontext nor Step1X-Edit v1.2 are reported in Tab. 1? Since they are the models used to create the training data for the proposed method it will be important to verify how they compare quality wise.

3. The generative model is only 900M parameters, but this method relies on a 2B LLM for instruction parsing. How does this compare wrt competing models? The parameters of the LLM are still part of the model and should be accounted for.

4. Can you clarify weakness B

**Details Of Ethics Concerns:**

Not too bad, but it would be nice to clarify: some of the figures in the paper are likelly covered by copyright, e.g., Fig. 20 is a frame from a movie. Fig. 19 seems to belong to getty images (results from reverse image search). The authors should have used open images for this test or at least clarified the source of their images, the same applies to all images in the paper.

---

### Note · Program_Chairs · 2025-11-12
**Submission Desk Rejected by Program Chairs**

The source code for the linked page (https://anoy1314.github.io) contains author identity and consequently this submission must be desk rejected.